# Macroscopic water vapor diffusion is not enhanced in snow

Kévin Fourteau[1], Florent Domine[2,3], and Pascal Hagenmuller[1]

[1]Univ. Grenoble Alpes, Université de Toulouse, Météo-France, CNRS, CNRM, Centre d'Études de la Neige, Grenoble, France
[2]Takuvik Joint International Laboratory, Université Laval (Canada) and CNRS-INSU (France), Québec, QC, G1V 0A6, Canada
[3]Centre d'Études Nordiques (CEN) and Department of Chemistry, Université Laval, Québec, QC, G1V 0A6, Canada

**Correspondence:** kfourteau@protonmail.com

**Abstract.** Water vapor transport in dry snowpacks plays a significant role for snow metamorphism and the mass and energy balance of snowpacks. The molecular diffusion of water vapor in the interstitial pores is usually considered as the main or only transport mechanism, and current detailed snow physics models therefore rely on the knowledge of the effective diffusion coefficient of water vapor in snow. Numerous previous studies have concluded that water vapor diffusion in snow is enhanced relative to that in air. Various field observations also indicate that for vapor transport in snow to be explained by diffusion alone, the effective diffusion coefficient should be larger than that in air. Here we show using theory and numerical simulations on idealized and measured snow microstructures that, although sublimation and deposition of water vapor onto snow crystal surfaces do enhance microscopic diffusion in the pore space, this effect is more than countered by the restriction of diffusion space due to ice. The interaction of water vapor with the ice results in water vapor diffusing more than inert molecules in snow, but still less than in free air, regardless of the value of the sticking coefficient of water molecules on ice. Our results imply that processes other than diffusion play a predominant role in water vapor transport in dry snowpacks.

## 1 Introduction

When a snowpack is submitted to a temperature gradient, macroscopic water vapor transfer occurs from the warmer to the colder parts of the snowpack, in a process sometimes referred to as layer-to-layer vapor flux. This redistribution of mass plays a significant role in the evolution of the snowpack and its physical properties. In the absence of air convection in the snowpack, this macroscopic vapor flux results from the microscopic vapor diffusion occurring in the interstitial pores of snow, and is impacted by water sublimation and deposition processes acting as sources and sinks of vapor at the ice-pore interface (Yosida et al., 1955; Colbeck, 1983). The physics at play in the pores is generally agreed upon, even though questions about the precise kinetics of the sublimation and deposition of water molecules onto ice surfaces in snow remain open (Legagneux and Domine, 2005; Pinzer et al., 2012; Calonne et al., 2014; Krol and Löwe, 2016). However, even for investigators assuming the same physics at the microscopic scale, the transition from the microscopic to the macroscopic scale remains a point of contention in the snow community (Giddings and LaChapelle, 1962; Colbeck, 1993; Pinzer et al., 2012; Hansen and Foslien, 2015; Shertzer and Adams, 2018; Hansen, 2019). Yet, a proper understanding of vapor transport in snow at the macroscopic scale is a pre-

requisite for accurate snowpack physical modeling.

There has notably been a long-standing controversy concerning the magnitude of the macroscopic diffusive fluxes transporting mass from one layer to another, and in particular to determine whether they are larger than what would be observed in free air under similar macroscopic vapor gradients. The seminal study of Yosida et al. (1955) set out to measure in the laboratory the macroscopic vapor flux in a pile of snow subjected to a thermal gradient. Their results indicated that contrary to first expectations, the vapor flux was about 3 to 4 times larger than in free air. To explain this enhanced diffusion, Yosida et al.

(1955) introduced the "hand-to-hand" delivery mechanism, which notably considers that the deposition of water molecules on one side of an ice grain and the sublimation on another side acts as a shortcut in the vapor trajectory. Several subsequent experimental studies have either confirmed (e.g. Sommerfeld et al., 1987) or contradicted (Sokratov and Maeno, 2000) the findings of Yosida et al. (1955) that macroscopic vapor diffusion is significantly larger in snowpacks than in free air. Similarly, several analytical and numerical modeling works have either accepted (Colbeck, 1993; Christon et al., 1994; Gavriliev, 2008;

Hansen and Foslien, 2015) or contradicted (Giddings and LaChapelle, 1962; Calonne et al., 2014; Shertzer and Adams, 2018) the results of Yosida et al. (1955) and the hand-to-hand mechanism. As mentioned by Sokratov and Maeno (2000) and Pinzer et al. (2012) the experimental discrepancies can be explained by the difficulty to accurately measure macroscopic vapor fluxes and vapor concentration gradients in snow, either in the field or in the laboratory. Yet, the large disagreement between the various analytical and modeling works, which sometimes differ more than tenfold (e.g., Colbeck, 1993; Calonne et al., 2014),

cannot be explained by experimental errors.

The aim of this paper is to clarify the origin of these discrepancies and to quantify the macroscopic vapor flux based on theoretical and numerical modeling. As the kinetics of sublimation and deposition of water molecules on the ice surfaces in snow is not well constrained, we decided to explore a broad range of possible kinetics in our study. We start by considering in Section 2 whether the hand-to-hand mechanism, as originally proposed by Yosida et al. (1955), can indeed explain the large

macroscopic vapor fluxes observed in snow. Then in Section 3, we recall how the macroscopic vapor flux can be obtained from the microscopic vapor flux occurring at the pore scale. In Section 4 we present theoretical work to bound the macroscopic vapor flux in snow, by treating two limiting cases of surface kinetics. Finally, numerical simulations are presented in Section 5 in order to illustrate the points raised throughout the article and to provide some numerical values of the effective diffusion coefficient.

## 2   Does the hand-to-hand mechanism enhance macroscopic vapor diffusion?

As previously mentioned, the experiment of Yosida et al. (1955) marks the introduction of the idea of enhanced vapor diffusion due to the hand-to-hand delivery mechanism. Their experimental set-up consisted of four stacked cans ($3.5\,\mathrm{cm}$ in height and $5.5\,\mathrm{cm}$ in diameter each) filled with snow, and separated with wire meshes that held the snow in place in each can without preventing vapor diffusion between them. A temperature difference was imposed between the top and bottom of the stack in

order to create a vertical thermal gradient of about $45\,\mathrm{K\,m^{-1}}$, and thus induce a macroscopic vapor flux. The experiments were carried out with average temperatures of about $-4\,^{\circ}\mathrm{C}$ and lasted about 5 hours. The cans filled with snow were weighed be-

fore and after the experiment in order to determine their mass gain or loss, which can be used to estimate the magnitude of the macroscopic vapor flux transporting mass from one can to another. Based on these measurements, and assuming that vapor was at saturation concentration, Yosida et al. (1955) concluded that the macroscopic vapor flux was about 3 to 4 times greater than what would be expected in free air for a similar concentration gradient. Noting that this result appears to contradict the idea that the presence of ice would impede the diffusion of vapor in snow, Yosida et al. (1955) proposed the hand-to-hand delivery mechanism as an explanation for this contradiction. This mechanism first states that because of its low thermal conductivity, the pore space of snow tends to concentrate the thermal gradient, leading to a concentrated vapor gradient in the pores. More-over, Yosida et al. (1955) proposed that: *"Water vapor needs not force its way through the interspaces between the ice grains composing snow. It needs only condense on one side of an ice grain and evaporate from the other side to condense again on the side facing to it of the next grain. In this way the distance which the water vapor actually traverses by diffusion turns out to be a fraction of the distance of its displacement. Such a situation makes the diffusion of water vapor through snow easier than through open air, which causes D [the effective diffusion coefficient in snow] to appear greater than $D_0$ [the diffusion coefficient in free air]"*. One should note that this explanation entails more than the simple continuous sublimation of vapor from some interfaces and subsequent deposition on others. Yosida et al. (1955) argued that this is equivalent to a situation in which a molecule depositing on one side of an ice grain re-appears as a sublimating molecule on another side.

Our understanding is however that the second part of the mechanism proposed by Yosida et al. (1955) is not physically sound, and that the continuous deposition and sublimation of molecules cannot be used to explain their experimental results. A schematic illustration of the experiment is given in Figure 1, with only two cans for simplicity. The hand-to-hand delivery of water molecules is represented by the orange and red dots, depositing on the lower side and sublimating on the upper side of the ice grain at the interface between the two cans. For this mechanism to explain the experimental observations, the continuous deposition and sublimation should produce a real mass flux from one can to the other, as if the depositing molecule reappeared as the sublimating one. However, what actually happens is that the depositing molecule (represented as an orange dot in Figure 1) remains incorporated at the bottom of the ice grain , thus remaining in the first can. Similarly, the sublimating molecule (represented as a red dot in Figure 1) was already present in the second can. The synchronous sublimation and deposition therefore do not lead to a mass transfer between the two cans. This is different from the molecules traversing the boundary in the air space (represented as green dots in Figure 1), that actually lead to a mass transfer by depleting the first can in favor of the second one. We therefore argue that the hand-to-hand mechanism, as proposed by Yosida et al. (1955), is not physically sound.

If one adopts the hand-to-hand mechanism, such as Hansen (2019) for instance, the idea of water vapor shortcutting the ice may appear supported by the indistinguishability of water molecules. For an observer focused on the pore space, the argument says, it really appears as if the water vapor is transported almost instantaneously through the ice, as a disappearing water molecule depositing on one side of an ice grain is almost instantaneously replaced by an appearing molecule sublimating on the other side. However, this point of view neglects the fact that the mass leaving a control volume also depends on the gain or loss of the ice during the deposition/sublimation process. As exemplified in the right panel of Figure 1, for an observer focused on the

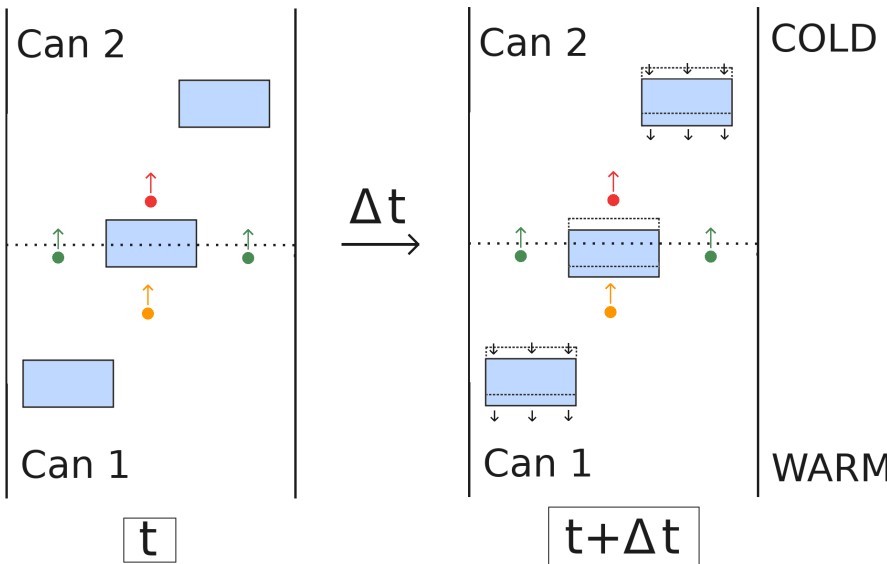

**Figure 1.** Illustration of the experiment of Yosida et al. (1955) (not to scale), with the ice space represented in blue and the boundary between two cans represented as a dashed line. The green dots represent water molecule diffusing through the boundary between two cans. The orange and red dots are depositing and sublimating molecules, which are at the origin of the hand to hand mechanism as proposed by Yosida et al. (1955). The evolution of the system over a time period $\Delta t$ is depicted in the right panel. The black arrows indicate the movement of the ice phase, opposite to that of water molecules in the air space.

ice everything appears as if the ice disappearing on the sublimation side reappeared on the depositing side (see for instance the videos in the Supplements of Pinzer et al., 2012; Hagenmuller et al., 2019). Because of mass conservation during the sublimation/deposition process, the apparent flux of vapor skipping the ice is compensated by an equal counter-flux of water

molecule in the ice space. Therefore, the mass transfer from one control volume to another is solely governed by the diffusion of water molecules in the air (green dots in Figure 1).

We stress that we do not disagree with the insightful propositions of Yosida et al. (1955) (i) that the vapor flux tends to travel from one ice grain to another and not to go around them, and (ii) that the thermal gradient is enhanced in the pore space compared to the macroscopic gradient. The point of contention is that the continuous sublimation and deposition of

100 water molecules does not count as a contribution to the mass flux. This problem with the hand-to-hand mechanism has been previously addressed by Giddings and LaChapelle (1962), when they noted that *"The hand-to-hand transfer does not contribute to the flux because this transfer does not shift water molecules across a plane fixed in the solid network"*.

The problem at hand is now to quantify the impact of the enhanced thermal gradient in the air space on the macroscopic diffusion of vapor, and to determine whether it can account for the large macroscopic vapor fluxes reported in the literature

(e.g. Yosida et al., 1955; Sommerfeld et al., 1987), and in particular if the macroscopic diffusion fluxes in snow are larger than the fluxes in free air.

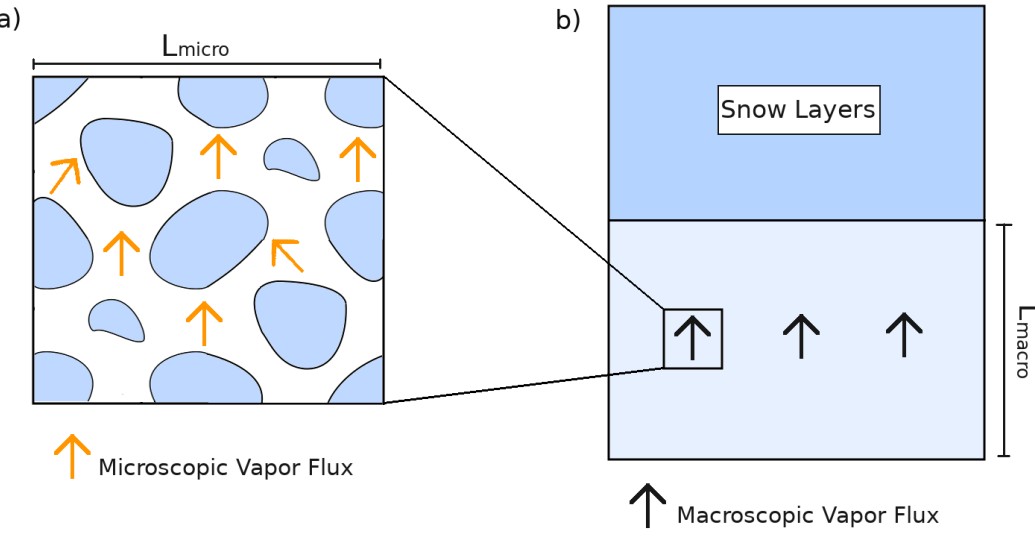

**Figure 2.** Relationship between the microscopic and macroscopic points of view of a snow sample. a) Microscopic point of view, with the ice in blue and microscopic vapor flux in orange. b) Macroscopic point of view where the snowpack is seen as a layered continuum.

## 3 Defining the macroscopic vapor flux and the effective diffusion coefficient

Let us consider a volume of snow (Figure 2a), subjected to vertical macroscopic temperature and vapor gradients at its boundaries. For this study we consider that the macroscopic water vapor gradient equals the macroscopic gradient of saturated vapor, and is therefore driven by the macroscopic temperature gradient (as in Yosida et al., 1955; Colbeck, 1993; Sokratov and Maeno, 2000; Pinzer et al., 2012). A necessary condition to be able to treat this snow sample as an equivalent macroscopic medium, is the condition of separation of scales (Auriault, 1991; Auriault et al., 2010). This separation of scale can be expressed as

$$L_{\mathrm{micro}} \ll L_{\mathrm{macro}} \tag{1}$$

where $L_{\mathrm{micro}}$ is the length-scale characterizing the size of the Representative Elementary Volume (REV) (Auriault et al., 2010; Calonne et al., 2014) of the microstructure, and $L_{\mathrm{macro}}$ is the length-scale characterizing variations of the snowpack or of the external forcing applied at the macroscopic scale, for instance the change between different snow layers or changes in thermal and vapor gradients (Figure 2b). In this study we consider snow samples with a size of at least $L_{\mathrm{micro}}$ but less than $L_{\mathrm{macro}}$. In this case, the snow sample is large enough to be treated as an equivalent macroscopic body, but no so large that it spans several snow layers and can thus be considered as macroscopically homogeneous. The relation between the various length-scales is exemplified in Figure 2.

At the microscopic scale, vapor diffuses in response to vapor concentration gradients in the pore space. The resulting microscopic vapor fluxes **f** are governed by Fick's law: $\mathbf{f} = -D_0 \nabla c$, with $D_0$ being the diffusion coefficient of vapor in air and $\nabla c$

the gradient of vapor concentration in the pore. These microscopic fluxes may result in a net transport of mass at the macroscopic scale, i.e. a macroscopic flux. The magnitude of this macroscopic flux $\mathbf{F}$ corresponds to the mass transported through an orthogonal plane per unit time and per unit surface of snow. This macroscopic flux is the quantity that Yosida et al. (1955) set out to measure.

This paper, as previous works in the scientific literature, will determine the macroscopic flux from the first principles of physics at the pore scale. It is therefore necessary to determine how the macroscopic flux $\mathbf{F}$ at the macroscopic scale can be obtained from the microscopic fluxes $\mathbf{f}$ in the pores. One might attempt to compute $\mathbf{F}$ as the quantity of matter transported through an arbitrary plane of the microstructure. In this case, $\mathbf{F}$ would be given as the surface average of the pore-scale flux $\mathbf{f}$, with the averaging performed over the entire plane, ice included (the vapor flux being zero in the ice). Yet, this method of computing the macroscopic vapor flux can be problematic. As pointed out by Pinzer et al. (2012) the water vapor fluxes through different horizontal planes of a microstructure are not necessarily all equal. Thus, depending on the plane chosen, the same snow sample could be assigned different macroscopic fluxes, contrary to the notion that the snow sample is homogeneous from the macroscopic point of view. To avoid this issue, the macroscopic flux should therefore be computed as the volume-averaged microscopic vapor flux over the entire representative volume of the microstructure (Shertzer and Adams, 2018), which is equivalent to averaging the fluxes through various horizontal planes (Pinzer et al., 2012). Again, the averaging needs to be performed over the total volume, including the ice space, and the macroscopic vapor flux $\mathbf{F}$ is thus given by

$$\mathbf{F} = \frac{1}{V} \int_{V_a} \mathbf{f} \, dV \tag{2}$$

where $V$ and $V_a$ respectively represent the total volume of the snow sample, and the pore volume.

We now phenomenologically define the effective diffusion coefficient for vapor $D_{\text{eff}}$ such that $\mathbf{F} = -D_{\text{eff}} \nabla C$, where $\nabla C$ is the macroscopic vapor concentration gradient (Colbeck, 1993; Shertzer and Adams, 2018). Here, the vapor concentration is expressed in mass per volume of pore space, and the averaging is thus performed in the pore only. The macroscopic vapor gradient is thus given by the difference in average vapor concentration between two opposing sides of the snow sample divided by the size of the sample. This corresponds to the definition implicitly adopted by Yosida et al. (1955). In the snow science community the effective diffusion coefficient $D_{\text{eff}}$ is usually expected to be independent of the applied thermal and vapor gradients (e.g. Yosida et al., 1955; Colbeck, 1993). In this case, it is possible to treat the problem of macroscopic vapor transport in snow with a generalized Fick's law, where $D_{\text{eff}}$ is independent of the applied boundary conditions and only depends on the snow microstructure. Such an effective diffusion coefficient does not depend on the external conditions, and is then said to be intrinsic (Auriault et al., 2010). However, one should keep in mind that the effective diffusion coefficients computed in this work might depend on the applied vapor and thermal gradients, and are therefore not necessarily intrinsic. Moreover the proposed numerical values may also not apply in the case where the macroscopic concentration gradient is decoupled from the macroscopic thermal gradient. Finally, we define the normalized effective diffusion coefficient as $D_{\text{eff}}^{\text{norm}} = D_{\text{eff}}/D_0$. Expressing macroscopic water vapor fluxes in snow under the form of normalized diffusion coefficients allows us to easily compare

them to free air.

Note that the goal of this work is only to quantify the macroscopic water vapor flux in snow and its associated phenomeno-
logical effective diffusion coefficient. Contrary to Calonne et al. (2014) we do not attempt to derive the macroscopic equations
governing water vapor at the layer scale.

## 4  Bounding the effective diffusion coefficient of water vapor in snow

Let us consider a snow sample of volume $V$ subjected to vertical thermal and vapor concentration gradients. For simplicity,
we assume the problem to be steady-state. The diffusion of water vapor at the microscopic scale is governed by the following
system of equations (Calonne et al., 2014)

$$
\begin{cases}
\mathrm{div}(-\mathrm{D}_0\nabla\mathrm{c}) = 0 & (\Omega_{\mathrm{a}}) \\
-\mathrm{D}_0\nabla c \cdot \mathbf{n} = \alpha v_{\mathrm{kin}}(c - c_{\mathrm{sat}}) & (\Gamma)
\end{cases}
\tag{3}
$$

where $\Omega_{\mathrm{a}}$, $\Gamma$, and $\mathbf{n}$ represent the pore space, the ice/pore interface, and the normal vector to $\Gamma$ pointing toward the ice. $D_0$ is
the vapor diffusion coefficient in free air, $c$ the vapor concentration in the pores, $c_{\mathrm{sat}}$ the vapor saturation concentration at the ice
interface, $v_{\mathrm{kin}} = \sqrt{(kT)/(2\pi m)}$ is related to the velocity of water molecules in the gas and is referred to as the kinetic velocity
($k$ being Boltzmann's constant and $m$ the mass of a water molecule), and $\alpha$ is the sticking coefficient of water molecules on the
ice surface (sometimes referred to as the accommodation coefficient), and is less than or equal to unity. The second equation
of the system is the Hertz-Knudsen equation and governs the mass fluxes that are incorporated or released from the ice. In the
presence of a large enough thermal gradient, the dependence of the saturation concentration to the local curvature of the ice
surface can be neglected compared to its dependence on temperature (Colbeck, 1983). Under this condition, we can expect $c_{\mathrm{sat}}$
to become a function of temperature only. Moreover, even if curvature effects were not negligible at the microscopic level it
appears unlikely for them to result in a net macroscopic vapor flux, as in a homogeneous snow layer curvature differences are
distributed isotropically within the microstructure, and thus do not result in a net movement of water vapor.

The actual value of the $\alpha$ coefficient is not well-known, and in general will depend on the local saturation of water vapor
and on the crystallographic properties of the ice surface (Saito, 1996; Libbrecht and Rickerby, 2013). Yet, two limiting cases,
corresponding to the case of infinitely fast surface kinetics and inert ice surfaces, can easily be analyzed. As will be empirically
verified later, these two cases appear to correspond to the upper and lower bounds of macroscopic vapor fluxes in snow. Solving
Equation 3 we obtain the microscopic vapor fluxes inside the whole microstructure. Using Equation 2 yields the water vapor
flux at the macroscopic scale $\mathbf{F}$.

## 4.1 The infinitely fast surface kinetics case

In the case where the product $\alpha v_{\mathrm{kin}}$ is very large, small oversaturations (or respectively undersaturations) lead to an abrupt adsorption (respectively desorption) of water molecules, rapidly restoring the saturation value. In the limiting, and hypothetical, case of infinitely fast surface kinetics (i.e. $\alpha v_{\mathrm{kin}} \to \infty$), the vapor concentration is constantly at saturation at the ice/pore interface and the Hertz-Knudsen equation can be replaced by the simpler equality of the vapor concentration with its saturation value at the ice surface. While the infinitely fast kinetics case is strictly theoretical, as $\alpha v_{\mathrm{kin}}$ is less than or equal to $v_{\mathrm{kin}}$, it

helps apprehending the macroscopic vapor flux when surface kinetics processes are much faster than diffusion in the air space. Note also, that the saturation of water vapor at the interface does not mean that the deposition and sublimation fluxes are zero at the interface.

   As explained by Pinzer et al. (2012), the infinitely fast surface kinetics situation is the case where the microscopic vapor gradients across the pores are maximal, and therefore where the macroscopic vapor flux is also maximal. A demonstration of this

fact, using the spatial averaging theorem is given in Appendix A. Note that the assumption of saturated vapor at the ice surface, and therefore infinitely fast surface kinetics, has been regularly employed in studies about the diffusion of vapor in snow (e.g. Colbeck, 1993; Christon et al., 1994; Pinzer et al., 2012).

   Even though this case corresponds to the maximal vapor flux, it can be shown that the macroscopic diffusion coefficient

remains less than expected in free air, as pointed out by Giddings and LaChapelle (1962). This is due to the loss of diffusion space because of the ice, and we propose here to rederive the Giddings and LaChapelle (1962) demonstration, using a more detailed framework. First, we assume that the thermal gradient is low enough, so that the saturation vapor concentration dependence on temperature can be considered to be linear. For a thermal gradient of $100 \, \mathrm{K \, m^{-1}}$ applied to a $1 \, \mathrm{cm}$ sample, the deviation of vapor concentration from linear behavior is about $0.1\%$, while the deviation of the derivative with respect

to temperature is about $5\%$. Moreover, this condition corresponds to the fact that the macroscopic vapor gradient should be constant over the sample, i.e. that the size of the sample is smaller than $L_{\mathrm{macro}}$.

   Under this assumption one can show that the vapor concentration is at saturation within the entire pore space. A demonstration is presented in Appendix B, and a similar conclusion was also reached by Yosida et al. (1955) and Pinzer et al. (2012). Consequently, the macroscopic vapor flux is expressed as

$$\mathbf{F} = \frac{1}{V} \int_{V_{\mathrm{a}}} \mathbf{f} \mathrm{d}V = \phi \frac{1}{V_{\mathrm{a}}} \int_{V_{\mathrm{a}}} -D_0 \nabla c_{\mathrm{sat}} \mathrm{d}V = \phi \frac{1}{V_{\mathrm{a}}} \int_{V_{\mathrm{a}}} -D_0 \frac{\mathrm{d}c_{\mathrm{sat}}}{\mathrm{d}T} \nabla T_{\mathrm{a}} \mathrm{d}V \tag{4}$$

where $\phi$ is the snow porosity (not to be confused with the ice volume fraction), $V_{\mathrm{a}}$ is the volume of the pore space, $\nabla T_{\mathrm{a}}$ is the microscopic temperature gradient in the air, and where we have used the chain rule $\nabla c_{\mathrm{sat}} = \frac{\mathrm{d}c_{\mathrm{sat}}}{\mathrm{d}T} \nabla T_{\mathrm{a}}$. As we considered

that the saturation concentration of vapor does not deviate from a linear behavior, $\frac{dc_{\text{sat}}}{dT}$ is taken as constant over the volume $V_a$. Thus

$$\mathbf{F} = -\phi D_0 \frac{dc_{\text{sat}}}{dT} \frac{1}{V_a} \int_{V_a} \nabla T_a dV \tag{5}$$

The precise relationship between the average microscopic thermal gradient in the air space, and the macroscopic gradient $\nabla T$ depends on the particular snow microstructure (Calonne et al., 2011, 2014; Hansen and Foslien, 2015). However, Hansen and Foslien (2015) report that

$$\nabla T = \phi \frac{1}{V_a} \int_{V_a} \nabla T_a dV + (1 - \phi) \frac{1}{V_i} \int_{V_i} \nabla T_i dV \tag{6}$$

where $V_i$ is the volume of the ice space and $\nabla T_i$ is the microscopic temperature gradient in the ice.

As snow is a transversely isotropic material with the vertical direction being the direction normal to the isotropy plane, one can expect for reason of symmetry that the average air and ice thermal gradients are aligned with the vertical macroscopic gradient. Moreover, the average air and ice thermal gradients are oriented in the same direction as the macroscopic gradient. Therefore, one has the inequality about the magnitudes of the air and macroscopic thermal gradients

$$\frac{1}{V_a} | \int_{V_a} \nabla T_a dV | \leq \frac{1}{\phi} |\nabla T| \tag{7}$$

which states that while the average thermal gradient in the air can be greater than the macroscopic thermal gradient, it cannot exceed it by a factor greater than $1/\phi$. Intuitively, it states that the temperature drop in the pore space cannot exceed the temperature drop observed over the entire snow sample. One can show that the air thermal gradient is maximal in the special case of a microstructure composed of slabs perpendicular to the macroscopic temperature gradient. In this case the temperature gradient is almost entirely concentrated in the air, and furthermore Equation 7 becomes an equality when the thermal conductivity of ice is assumed to be infinite.

Using the inequality of Equation 7 in Equation 5, leads to an inequality on the magnitude of the macroscopic flux

$$|\mathbf{F}| \leq D_0 \frac{dc_{\text{sat}}}{dT} |\nabla T| = D_0 |\nabla C| \tag{8}$$

where $\nabla C = \frac{dc_{\text{sat}}}{dT} \nabla T$ is the macroscopic vapor concentration gradient.

The macroscopic vapor flux is thus less than the vapor flux that would take place in free air, which can be similarly expressed by $D_{\text{eff}}^{\text{norm}} \leq 1$. While the microscopic vapor flux in the pores is enhanced due to the enhancement of the microscopic temperature and vapor gradients, this effect is countered by the reduction of the space where vapor can diffuse. As the average air

temperature gradient is at the maximum enhanced by a factor $1/\phi$ while the reduction of pore space systematically decrease the macroscopic flux by a factor $\phi$, the resulting macroscopic vapor flux cannot be greater than in free air. The equality $D_{\text{eff}}^{\text{norm}} = 1$ holds when the entire temperature gradient is concentrated in the pore space. However, since the thermal conductivity of ice is finite, the thermal gradient cannot be solely concentrated in the pore space and thus one always has $D_{\text{eff}}^{\text{norm}} < 1$.

## 4.2 The slow surface kinetics case

The other limiting case is when the deposition and sublimation of water vapor at the ice grain surfaces is slow enough to be neglected. The diffusion of water vapor in snow then becomes equivalent to the diffusion of a gas in an inert porous structure. This problem has been extensively studied (e.g. Torquato and Haslach Jr, 2002; Auriault et al., 2010), and in this case the effective diffusion coefficient is given by

$$D_{\text{eff}} = \phi \tau D_0 \tag{9}$$

where $\tau$ is defined as the tortuosity factor and is linked to the lengthening of the diffusion streamlines in the porous network. The tortuosity factor represents an impediment of diffusion, and is thus less than or equal to unity. Moreover, $\tau$ depends solely on the structure of the porous medium and not on the specific diffusive specie or the applied concentration gradient (Torquato and Haslach Jr, 2002; Auriault et al., 2010). Under an assumption of slow surface kinetics, Calonne et al. (2014) report effective diffusion coefficients reduced from 20 to $85\%$ compared to the free air case, with lower diffusion coefficients corresponding to denser snow samples. Although we do not have a rigorous demonstration of this fact, it appears that the slow kinetics assumption corresponds to the case where the macroscopic flux (and hence $D_{\text{eff}}$) is minimal for a given vapor concentration gradient. This proposition will be empirically verified with numerical simulations in Section 5.

## 4.3 Comparison with previous works

We have established in Section 4.1 that even under the assumption of fast surface kinetics, the effective vapor diffusion coefficient in snow cannot be greater than that in free air. Yet several studies based on analytical and numerical models, which are not subjected to experimental errors, have reported opposite results. It thus appears important to elucidate why those previous results do not invalidate the demonstration made in Section 4.1 and the results of this work.

Colbeck (1993) proposed a theoretical model, based on an idealized structure of disconnected and equally spaced ice spheres. In that model the vapor concentration is at saturation at the ice surface (i.e. surface kinetics are infinitely fast) and the vapor flux between two consecutive spheres can be analytically computed. In this case, the author concludes that the vapor diffusion coefficient is between four to seven times greater than in air. However, as pointed out by Pinzer et al. (2012), Colbeck (1993) derives the diffusion coefficient in snow by computing the flux crossing a single plane between two spheres, and not by averaging over the entire volume. As the plane between two spheres corresponds to a zone of maximal thermal gradient without

any ice blockage, it is not surprising that the local microscopic vapor flux is several-fold that in free air. However, as will be seen in Section 5.1, computing the macroscopic flux by performing a volume averaging of microscopic vapor fluxes over the entire microstructure significantly reduces the corresponding diffusion coefficient, down to a value below that of free air.

Christon et al. (1994) performed finite element microscale simulations of vapor diffusion in snow under a thermal gradient, using an idealized microstructure. They concluded that the vapor diffusion coefficient is between one and two times as large

as that in air. Yet, in that study the macroscopic mass flux is not computed as a volume average, but rather *"as the weighted average of the mass flux rates over all of the exterior surfaces of the diffusion domain in order to capture the bulk vertical mass diffusion rate"*. Here, the diffusion domain refers to the domain where vapor diffusion occurs, i.e. the pore space. This differs from volume averaging and leads to an overestimation of the macroscopic flux, as the ice space is not included. As the loss of diffusion space due to the ice is neglected, the effective diffusion coefficient is overestimated by a factor of $1/\phi$.

Similarly, Pinzer et al. (2012) performed finite element microscale simulations of vapor diffusion, this time with microstructures measured by X-ray computed microtomography scanning. A diffusion coefficient slightly greater than that in free air is reported. Pinzer et al. (2012) noted that computing the mass flux crossing a single plane was insufficient, for the reasons discussed in Section 3. To derive the macroscopic mass flux, Pinzer et al. (2012) computed the average mass flux in each plane, and then averaged over all planes. However, it appears from the description of their methodology that the slice averaging was

only performed in the pore space, not taking into account the reduction of macroscopic flux due to the presence of ice. As in the case of Christon et al. (1994), this would explain the diffusion coefficient higher than in free air. As will be shown in Section 5, performing similar numerical simulations and computing the macroscale flux by total volume averaging leads to diffusion coefficients below that in free air.

Finally, Hansen and Foslien (2015) and Hansen (2019) proposed an analytical expression for the effective thermal conduc-

290 tivity of snow, taking into account the latent heat associated with the transport of water vapor. In their model, water vapor is at constant saturation in the pores (thus corresponding to the case of infinitely fast kinetics), and acts an integral part of heat transfer by transporting latent heat between sublimation and deposition surfaces (as notably proposed by Yosida et al., 1955). One application of this effective thermal conductivity model is to allow the derivation of the vapor flux, which leads to the conclusion that the macroscopic vapor flux is greater than that in free air. To come to this conclusion, Hansen and

295 Foslien (2015) determine the vapor flux by identifying the contribution of latent heat in their expression of the effective thermal conductivity. However, during the identification of the latent heat contribution to the total energy flux, some of the heat conduction contribution of the ice is attributed to the latent heat transport. This leads to an artificially increased vapor flux, and therefore an overestimated diffusion coefficient. A re-derivation of the vapor flux with the thermal conductivity expression proposed by Hansen and Foslien (2015) is presented in Appendix C and leads to a macroscopic vapor flux below that in free air.

Most of the discrepancies between our results and those of the published literature thus reduce down to computations of the macroscopic fluxes that are inconsistent with fluxes expressed per unit surface of snow, as used in snow models and experimental studies. This leads to an overestimation of the value of the effective diffusion coefficient. Focusing on the magnitude of microscopic vapor fluxes as done by Colbeck (1993) or Christon et al. (1994) is of a great interest for snow metamorphism, as

they govern the mass transfer between adjacent ice grains and the recrystallization rate. However, they do not correspond to the macroscopic mass flux expressed per unit surface of snow, as measured by Yosida et al. (1955) and subsequent experimental studies (e.g. Sokratov and Maeno, 2000). We reiterate that the macroscopic vapor flux responsible for the redistribution of mass at the macroscopic scale, and which inspired the hand-to-hand delivery mechanism, corresponds to the volume-averaged flux over the entire snow microstructure and must include the loss of diffusion space due to the ice.

## 5 Numerical modeling

In this section we present steady-state 3D numerical simulations of vapor diffusion in snow subjected to a macroscopic temperature gradient $\nabla T$ and a macroscopic vapor gradient $\nabla C$. The macroscopic temperature gradient $\nabla T$ is obtained by imposing the top and bottom temperatures $T^{\text{top}}$ and $T^{\text{bot}}$. The vapor concentrations in the pore space at the top and bottom of the sample are imposed to correspond to the saturation values for the top and bottom temperatures. We thus have

$$|\nabla C| = \frac{|c_{\text{sat}}(T^{\text{top}}) - c_{\text{sat}}(T^{\text{bot}})|}{L_{\text{z}}} \tag{10}$$

where $L_{\text{z}}$ is the height of the sample considered. Conditions of zero heat and vapor normal fluxes are imposed on the other sides of the sample. For simplicity, we only consider the case of vertical temperature and vapor gradients, although the extension to the other directions is straight forward. Moreover, we do not take into account the impact of latent heat on the temperature field. At the microscopic level, adding latent heat effects would act as an additional mechanism transporting heat from the warm sublimating surfaces towards the cold deposition surfaces. It would cool the sublimation surfaces and warms the deposition surfaces, decreasing the thermal gradient in the pore space. Therefore, taking latent heat effects into account would not increase the effective vapor diffusion coefficient.

The thermal conductivities of the ice and the air $k_{\text{i}}$ and $k$a are set to 2.34 and $0.024 \, \text{W} \, \text{K}^{-1} \, \text{m}^{-1}$ respectively (Riche and Schneebeli, 2013), and the diffusion coefficient of vapor in air $D_0$ is set to $2 \times 10^{-5} \, \text{m}^2 \, \text{s}^{-1}$ (Calonne et al., 2014). The vapor concentration is assumed to follow the Clausius-Clapeyron and ideal gas laws, leading to

$$c_{\text{sat}} = \frac{M}{RT} P_0 \, \text{e}^{\left(\frac{\Delta H_{\text{s}}}{R}\left(\frac{1}{T_0} - \frac{1}{T}\right)\right)} \tag{11}$$

where $M = 18 \times 10^{-3} \, \text{kg} \, \text{mol}^{-1}$ is the molar mass of water, $R = 8.314 \, \text{J} \, \text{K}^{-1} \, \text{mol}^{-1}$ is the ideal gas constant, $\Delta H_{\text{s}} = 51 \times 10^3 \, \text{J} \, \text{mol}^{-1}$ is the latent heat of sublimation of ice, $T_0 = 273.15 \, \text{K}$ is a reference temperature, and $P_0 = 611 \, \text{Pa}$ is the saturation pressure of vapor over ice at $T_0$. The different physical constants used in this article are tabulated in Appendix D with their references. All simulations are performed with an average temperature $(T^{\text{bot}} + T^{\text{top}})/2 = 258 \, \text{K}$.

The heat and diffusion equations are solved using the finite element method with the open-source software ElmerFEM (Malinen and Råback, 2013). We use the readily available ElmerFEM modules dedicated to the heat and diffusion equations, which are solved with iterative methods. We first solve the steady-state heat equation in order to obtain the temperature field in the entire microstructure. The steady-state vapor diffusion equation is then solved using the saturation concentration at the ice/pore interface resulting from the previously computed temperature field. In the case of simulations performed on measured snow

microstructures, the tetrahedral meshes have been derived from Xray computed microtomography images using the CGAL meshing library. The meshes have been refined to capture the ice/pore interface, and contains between 18 and 50 million elements, depending on the snow sample. Moreover, in the case of snow samples the meshes have been partitioned into 20 sub-meshes and the computations are performed using the parallel computing abilities of ElmerFEM. Under such conditions, a
simulation typically takes a bit less than an hour to run. Finally, the outputs of the simulations are processed using the ParaView software to compute the volume averages.

As seen previously, the kinetics of the sublimation and deposition processes at the ice surface might significantly impact the magnitude of the macroscopic vapor flux. We recall that in general the boundary condition at the ice/air interface is given by
the Hertz-Knudsen equation

$$-D_0 \nabla c \cdot \mathbf{n} = \alpha v_{\mathrm{kin}}(c - c_{\mathrm{sat}}) \tag{12}$$

where $v_{\mathrm{kin}} \simeq 140\,\mathrm{m\,s^{-1}}$ at $258\,\mathrm{K}$, and $\alpha$ is the sticking coefficient less than or equal to unity. In general $\alpha$ is not a constant and depends on the local vapor saturation as well as the crystallographic properties of the underlying ice crystal (Saito, 1996; Libbrecht and Rickerby, 2013).
For each microstructure, several simulations were performed with different values of $\alpha$ in order to assess the impact of the internal boundary conditions (IBC) applied at the ice surface. We first performed simulations with constant $\alpha$ equal to 0, $10^{-5}$, $10^{-4}$, $10^{-3}$, $10^{-2}$, $10^{-1}$, and 1. Simulations with constant $\alpha$ are referred to as linear kinetics simulations in what follow. Among them, a special case is $\alpha = 0$ which corresponds to the diffusion of vapor in an inert porous medium. Moreover, we performed simulations similar to those of Christon et al. (1994) and Pinzer et al. (2012) where the Hertz-Knudsen boundary
condition is replaced with the saturation of vapor at the ice surface, corresponding to the infinitely fast kinetics case. Finally, we performed simulations in which the dependence of $\alpha$ to the local vapor saturation is explicitly represented. For that we set $\alpha = \exp(-\sigma_0/\sigma)$ where $\sigma = (c - c_{\mathrm{sat}})/c_{\mathrm{sat}}$ and $\sigma_0 = 0.01$. Note that this expression was determined for the attachment of vapor to the basal and prismatic facets of ice crystals (Libbrecht and Rickerby, 2013), and might not properly apply for the entirety of ice surfaces in snowpacks. Indeed, this law has been derived using deposition measurement, and might not apply
for sublimating surfaces (Beckmann and Lacmann, 1982). Moreover, the presence of vicinal surfaces in snowpacks, where the proposed law does not apply, is likely (Legagneux and Domine, 2005). Therefore, the point of using such a law is to qualitatively study the potential impact of a dependence of $\alpha$ to the local vapor saturation, rather than to produce quantitative results. Simulations using this law are referred to as non-linear kinetics simulations. Finally, the macroscopic fluxes of the various simulations are computed by performing a total volume average, as defined in Section 3, and the effective diffusion coefficients
are obtained by dividing these macroscopic fluxes by the macroscopic concentration gradients, i.e. $D_{\mathrm{eff}} = -\mathbf{F}/\nabla C$.

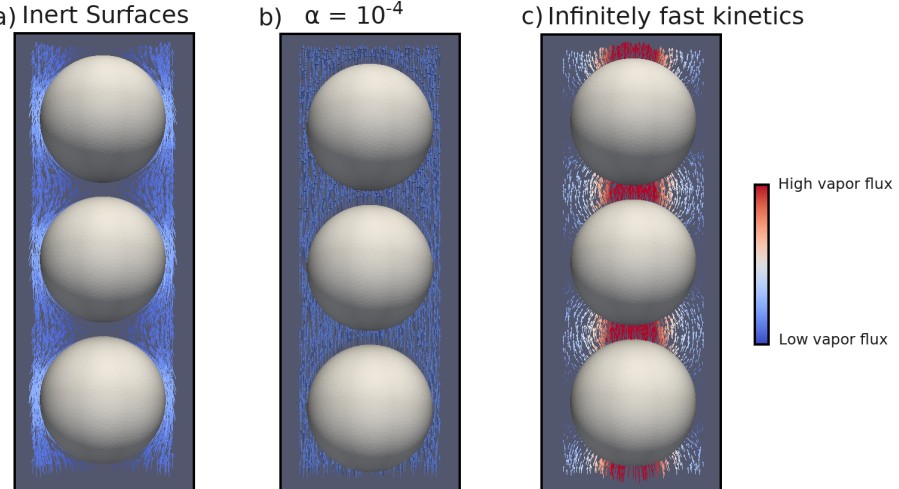

**Figure 3.** Disconnected ice spheres geometry with microscopic vapor fluxes in the pore space and for a $50\,\mathrm{K\,m^{-1}}$ thermal gradient. a) Inert surfaces case, b) $\alpha = 10^{-4}$ case, and c) infinitely fast kinetics case.

## 5.1 Idealized structure

We start with an idealized microstructure composed of disconnected ice spheres, similar to that used by Colbeck (1993). The structure is visible in Figure 3. The domain is a cuboid of dimension $3.7 \times 3.7 \times 10\,\mathrm{mm^3}$, with three equidistant ice spheres with 3 mm diameters and which are vertically aligned at the center of the domain. The distance between two sphere centers is set to 3.3 mm. This microstructure is characterized by a porosity of $0.619$ and a density of $349\,\mathrm{kg\,m^{-3}}$.

The simulations were performed for the different IBCs described previously and for temperature gradients ranging from 5 to $200\,\mathrm{K\,m^{-1}}$. The resulting normalized effective diffusion coefficients are displayed in Figure 4.

We first analyze the $50\,\mathrm{K\,m^{-1}}$ temperature gradient simulations. Illustrations of the microscopic vapor fluxes for three IBCs, namely inert surfaces ($\alpha = 0$), $\alpha = 10^{-4}$ and infinitely fast surface kinetics, are displayed in Figure 3. In the case of inert surfaces the vapor flux needs to go around the ice grains, which act as a blockage, leading to tortuous stream lines. In the case of infinitely fast surface kinetics, the vapor flux does not need to go around the ice grain and is rather moving from ice grain to ice grain, in agreement with the suggestion of Yosida et al. (1955) and the numerical simulations of Pinzer et al. (2012). Finally, the $\alpha = 10^{-4}$ case displays an intermediate behavior, with some of the vapor flux moving from ice grain to ice grain, while the rest bypasses the ice. This exemplifies that the microscopic vapor fluxes are strongly dependent on the kinetics of the vapor sublimation and deposition at the ice surface.

In the case of infinitely fast surface kinetics we find a normalized diffusion coefficient of $0.978$, i.e. lower than in air, in agreement with the calculations of Section 4.1. Moreover, we computed the average air temperature gradient (in the pore space only), and found it to be $79.00\,\mathrm{K\,m^{-1}}$. This is enhanced compared to the $50\,\mathrm{K\,m^{-1}}$ macroscopic gradient, but still respects the

inequality of Equation 7. While the enhancement of the thermal gradient increases the microscopic vapor fluxes in the pores, it does not suffice to counter the loss of diffusion space, and the resulting macroscopic flux is lower than in free air.

To compare our results to the works of Colbeck (1993), Christon et al. (1994), and Pinzer et al. (2012), who worked under the similar assumption of infinitely fast kinetics, we used two alternate methods, different from total volume averaging, to compute the vapor flux. The first consists in averaging the microscopic vapor fluxes in the air space only, and we call the associated normalized diffusion coefficient $D_{\text{air}}^{\text{norm}}$. The second one consists in computing the flux crossing an horizontal plane placed between two spheres, and we call the associated diffusion coefficient $D_{\text{plane}}^{\text{norm}}$. As explained in Section 4.3, we believe that the first methodology is akin to works of Christon et al. (1994) and Pinzer et al. (2012), while the second was used by Colbeck (1993). Calculations yield a $D_{\text{air}}^{\text{norm}}$ of 1.580 and a $D_{\text{plane}}^{\text{norm}}$ of 2.986, consistent with the values reported by Christon et al. (1994), Pinzer et al. (2012), and Colbeck (1993). By not including the ice in the averaging or by selecting a peculiar plane where microscopic vapor fluxes are maximum, the macroscopic vapor flux is overestimated, leading to a diffusion coefficient greater than $D_0$. The outcome of the other simulations performed with $\nabla T = 50\,\text{K}\,\text{m}^{-1}$ is reported in Figure 4 and indicates that $D_{\text{eff}}^{\text{norm}}$ is maximal in the infinitely fast kinetics case, with a value of 0.978, and minimal in the inert surfaces case, with a value of 0.512. Accordingly, the normalized effective diffusion coefficient increases with $\alpha$, and for the cases $\alpha = 0.1$ and $\alpha = 1$ differs by less than 0.3% from the infinitely fast case. The use of the non-linear surface kinetics law leads to a normalized effective diffusion coefficient equals to 0.857, in between the inert ($D_{\text{eff}}^{\text{norm}} = 0.512$) and infinitely fast kinetics ($D_{\text{eff}}^{\text{norm}} = 0.978$) cases.

Similar observations can be made for the simulations performed with other temperature gradients. For the entire range of gradients tested, the infinitely fast kinetics and inert surfaces cases correspond to the maximal and minimal macroscopic fluxes. Moreover, the associated effective diffusion coefficients are mostly independent of the macroscopic thermal or vapor gradients, suggesting that the effective diffusion coefficients could be intrinsic in these cases. Consistent results are observed for the simulations where $\alpha$ is constant. The obtained effective diffusion coefficients are mostly independent of the applied macroscopic gradient, and are bounded by the infinitely fast kinetics and inert surfaces cases. Note that the $\alpha = 0.1$ and $\alpha = 1$ cases are indistinguishable from the infinitely fast kinetics results in Figure 4. Contrary to the rest of the simulations, the non-linear IBC yields effective diffusion coefficients that depend on the magnitude of the applied gradients. In this case, the macroscopic vapor flux and the macroscopic vapor concentration gradient are not proportionally linked by a single and well-defined material property. Furthermore, for low vapor and thermal gradients the non-linear case is close to the inert surfaces case while a transition towards the fast kinetics case is observed for thermal gradients around $50\,\text{K}\,\text{m}^{-1}$. Again, even though the non-linear law used to express $\alpha$ as a function of local saturation does not necessarily accurately model water molecule attachment in real snowpacks, it illustrates the effects of a non-constant $\alpha$.

## 5.2 Measured snow microstructures

Other numerical simulations of vapor diffusion have been performed, this time using measured snow microstructures instead of the idealized structure of Section 5.1. The microstructures were obtained by X-ray computed microtomography imaging of snow samples. In total 6 snow samples were analyzed, covering the snow types of decomposing and fragmented precipitation

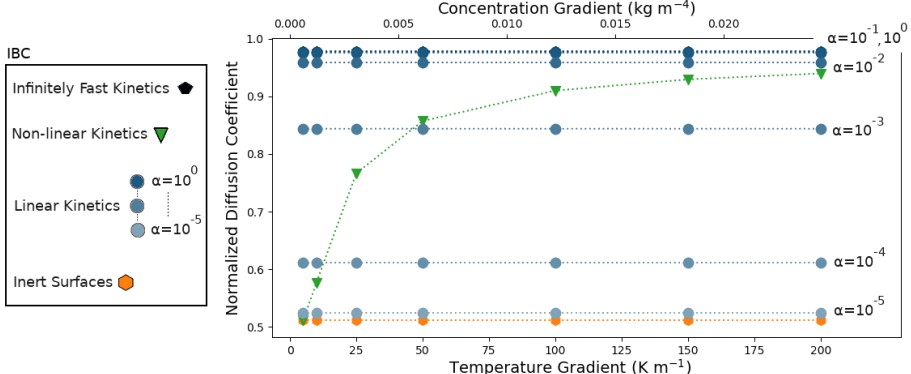

**Figure 4.** Normalized diffusion coefficients $D_{\mathrm{eff}}^{\mathrm{norm}}$ in the idealized spheres microstructure, for different temperature/vapor gradients, different IBCs and a mean temperature of $258\,\mathrm{K}$. Note that the $\alpha = 0.1$, $\alpha = 1$, and infinitely fast kinetics cases are indistinguishable at the top of the graph.

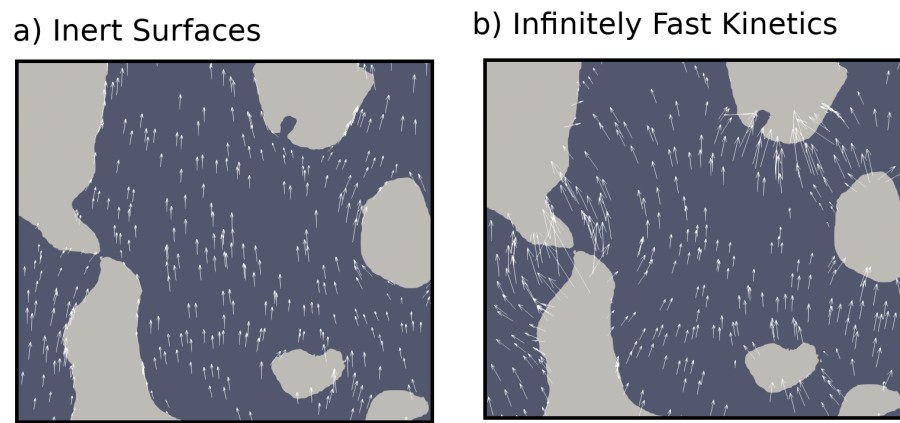

**Figure 5.** Vapor streamlines inside the melt forms sample, for a temperature gradient of $50\,\mathrm{K\,m^{-1}}$ and the inert surfaces and infinitely fast kinetics cases. Note that the arrows showing the vapor flux are centered around the point they represent, and might therefore wrongly appear to originate from or terminate in the ice.

particles (DF), depth hoar (DH), rounded grains (RG), and melt forms (MF) (Fierz et al., 2009). The goal is not to provide effective diffusion coefficients on an exhaustive set of snow microstructural patterns but to illustrate the effects of the snow microstructure and surface kinetics on water vapor diffusion.

A close-up view showing the vapor stream lines inside the melt forms sample is provided in Figure 5. As with the idealized microstructure, in the inert surface case vapor tends to go around the ice grains. In the infinitely fast kinetics case, vapor moves

from ice grain to ice grain, as proposed by Yosida et al. (1955) and reported by Pinzer et al. (2012).

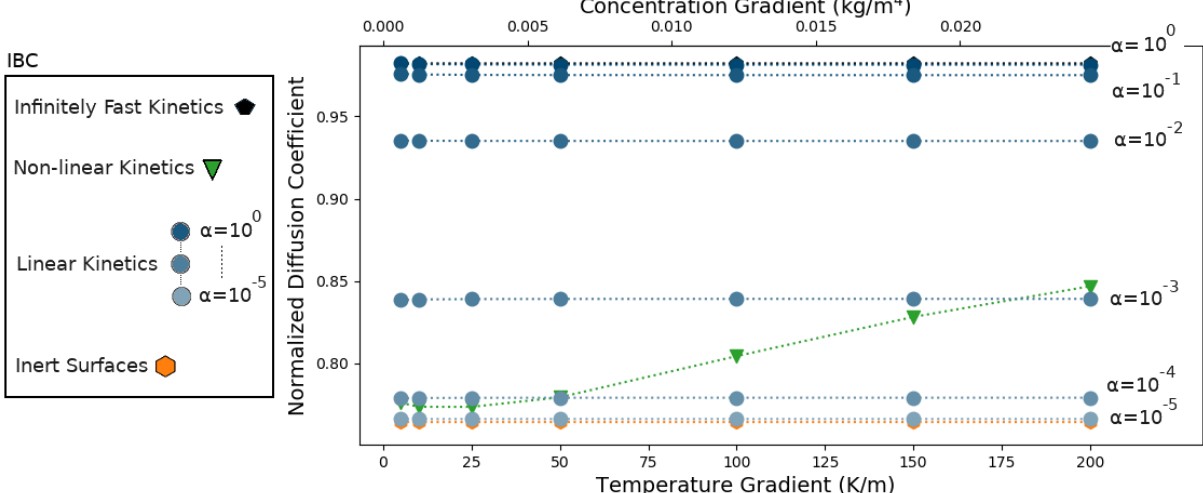

**Figure 6.** Normalized diffusion coefficients $D_{\text{eff}}^{\text{norm}}$ with a DF snow microstructure, for different temperature/vapor gradients, different IBCs and a mean temperature of $258\,\text{K}$. Note that the $\alpha = 1$ and the infinitely fast kinetics cases are superimposed at the top of the graph, and that the $\alpha = 10^{-5}$ and the inert surface cases are indistinguishable at the bottom of the graph.

We start by analyzing the results of the simulations of the DF sample, characterized a density of $125\,\text{kg}\,\text{m}^{-3}$. Similarly to Section 5.1, the simulations were performed by imposing external temperature and vapor gradients, with different selected IBCs characterizing the kinetics of the vapor sublimation and deposition process. The results are displayed in Figure 6. As in

the idealized case, the inert surface, infinitely fast kinetics, and linear kinetics cases yield normalized effective diffusion coefficients that are mostly independent of the applied gradients. Moreover, we observe that $D_{\text{eff}}^{\text{norm}}$ is minimal in the inert surface case with a value of $0.764$, and maximal in the infinitely fast kinetics case with a value of $0.982$. As expected, the effective diffusion coefficient is systematically lower than that of air. The normalized effective diffusion coefficients in the linear kinetics cases are distributed between the inert and infinitely fast values, and increase with the value of $\alpha$. For $\alpha = 1$, $D_{\text{eff}}^{\text{norm}}$ differs by

less than $0.1\%$ from the infinitely fast kinetics case.

On the contrary, the non-linear kinetics case leads to a normalized effective diffusion coefficient that depends on the external gradients. As with the idealized disconnected spheres structure of Section 5.1, we observe that for low gradients the non-linear case is close to the slow kinetics simulations, and transitions towards faster kinetics with higher gradients. However, in the case of the DF sample this transition occurs more slowly and with higher temperature and vapor gradients.

Since the normalized effective diffusion coefficients appear to be independent of the external thermal/vapor gradient in the case of infinitely fast and linear surface kinetics, we only computed $D_{\text{eff}}^{\text{norm}}$ with a $50\,\text{K}\,\text{m}^{-1}$ gradient for the 5 remaining snow samples. We also did not compute $D_{\text{eff}}^{\text{norm}}$ with non-linear surface kinetics (i.e. when alpha is not constant), as we are not confident in the validity of the chosen non-linear law for snow modeling. The resulting $D_{\text{eff}}^{\text{norm}}$ values are reported in Table 1,

**Table 1.** Computed normalized effective diffusion coefficients as a function of surface kinetics (columns) and snow sample (lines). Values are derived from simulations with a $50\,\mathrm{K\,m^{-1}}$ thermal gradient, but our results suggest that they are independent of the thermal gradient. Snow types are classified according to Fierz et al. (2009) and SSA stands for Specific Surface Area.

| Snow characteristics | | | $D_{\mathrm{eff}}^{\mathrm{norm}}$ | | | | | | | |
|---|---|---|---|---|---|---|---|---|---|---|
| Snow Type | Density $(\mathrm{kg\,m^{-3}})$ | SSA $(\mathrm{m^2\,kg^{-1}})$ | Inf. Fast Kinetics | $\alpha = 1$ | $\alpha = 10^{-1}$ | $\alpha = 10^{-2}$ | $\alpha = 10^{-3}$ | $\alpha = 10^{-4}$ | $\alpha = 10^{-5}$ | $\alpha = 0$ |
| DF | 125 | 40 | 0.982 | 0.981 | 0.975 | 0.935 | 0.839 | 0.779 | 0.766 | 0.764 |
| DH | 145 | 29 | 0.982 | 0.982 | 0.977 | 0.943 | 0.841 | 0.763 | 0.744 | 0.741 |
| DH | 156 | 26 | 0.977 | 0.977 | 0.973 | 0.942 | 0.840 | 0.744 | 0.718 | 0.714 |
| DH | 177 | 18 | 0.963 | 0.963 | 0.960 | 0.937 | 0.845 | 0.723 | 0.674 | 0.665 |
| RG | 316 | 34 | 0.913 | 0.910 | 0.894 | 0.807 | 0.646 | 0.561 | 0.539 | 0.532 |
| MF | 380 | 5 | 0.796 | 0.796 | 0.795 | 0.779 | 0.690 | 0.538 | 0.466 | 0.450 |

and displayed in Figure 7 as a function of the sticking coefficient $\alpha$. Again, $D_{\mathrm{eff}}^{\mathrm{norm}}$ is systematically minimal in the inert surface case and maximal in the infinitely fast kinetics. Figure 7 highlights that the normalized effective diffusion coefficient exhibits two different regimes depending on the value of $\alpha$. The transition between the fast and slow surface kinetics regimes occurs for values of $\alpha$ around $10^{-3}$.

We observe that the effective diffusion coefficient is well correlated with density, and show an almost systematic decrease of $D_{\mathrm{eff}}^{\mathrm{norm}}$ with increasing density, for all values of $\alpha$. The correlation between $D_{\mathrm{eff}}^{\mathrm{norm}}$ and the specific surface area is not so well marked, notably for the RG sample that shows a large value of specific surface area without any clear impact on $D_{\mathrm{eff}}^{\mathrm{norm}}$. That being said, our sample set is only composed of six samples and for which density and specific surface area are correlated. A detailed study of the influence of microstructural parameters on the effective diffusion coefficient would require a larger sample set, notably to be able to decipher the independent influence of specific surface area and density.

## 6 Discussion

We have shown that the macroscopic vapor flux in snow is less than the flux in free air under the same water vapor gradient. This result is supported by a formal demonstration, inspired by the work of Giddings and LaChapelle (1962), as well as by numerical simulations on idealized and measured snow microstructures. While the interaction of water vapor with the ice structure results in a macroscopic flux larger than that of the inert diffusion case, the macroscopic vapor flux cannot be enhanced compared to the free air case. We have shown that most of the previous theoretical studies reporting macroscopic vapor flux enhanced

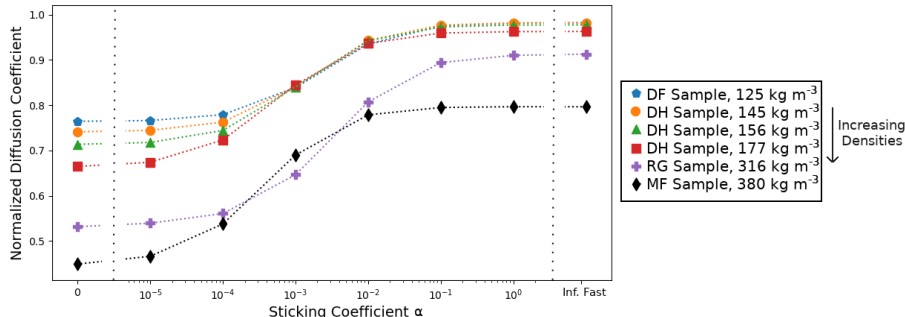

**Figure 7.** Normalized diffusion coefficients $D_{\text{eff}}^{\text{norm}}$ as a function the sticking coefficient $\alpha$, for the 6 snow samples considered in this paper.

compared to free air used faulty computations of the macroscopic vapor flux, which resulted in systematic overestimation.

As seen in this work, the sublimation/deposition fluxes at the ice surface play a great role on the final macroscopic flux. In particular we have shown that when the reaction is fast, i.e. $\alpha$ is large, the macroscopic fluxes can be close to those that would be observed in free air. Moreover, the dependence of $\alpha$ on the local vapor saturation might break the proportionality between the macroscopic vapor gradient and the macroscopic flux. In this case, it is no longer possible to define a single effective diffusion coefficient $D_{\text{eff}}$ that proportionally relates the vapor flux to vapor gradient, and that solely depends on the snow microstructure. In other words, with non-linear surface kinetics $D_{\text{eff}}$ is not intrinsic. For all these reasons, it appears important to determine what are the precise internal boundary conditions that govern the sublimation and deposition of water vapor in snowpacks, and in particular to determine whether the inert surfaces or infinitely fast kinetics case could accurately describe real snow. In the case of fast kinetics, one can have $D_{\text{eff}} \geq \phi D_0$, as the average microscopic vapor gradient can be greater than the macroscopic vapor gradient. On the contrary, in the case of slow surface kinetics one has $D_{\text{eff}} = \phi \tau D_0 \leq \phi D_0$, since $\tau \leq 1$. An experimental distinction between fast and slow kinetics could thus be made by observing whether the quantity $D_{\text{eff}}/(\phi D_0)$ is greater than unity or not. Using the experimental results of Sokratov and Maeno (2000), which are the experimental results with the lowest reported diffusion coefficient, we observe that $D_{\text{eff}}/(\phi D_0)$ is almost always greater than unity, which supports the notion of fast rather than slow kinetics. This is consistent with the study of Krol and Löwe (2016) which report that fast kinetics is consistent with their microtomography-based observation of the temperature gradient metamorphism of a snow sample. That being said, experimental determination of the macroscopic vapor fluxes is difficult, as exemplified by the large spread of reported values, and more observations would be needed to decisively conclude on this point.

This work investigated the effective diffusion coefficient of vapor in snow with a phenomenological approach, where the diffusion coefficient is simply defined as the ratio of the macroscopic vapor flux to the vapor concentration gradient. A rigourous upscaling of the microscale equations to derive the equivalent macroscopic formulation would greatly benefit the understanding and modeling of the macroscopic vapor flux. Note that such an approach was used by Calonne et al. (2014) with the method of asymptotic-scale expansion, but limited itself to small $\alpha$. Applying a similar method to the case of non-negligible surface sublimation and deposition would lead to a proper definition of the macroscopic quantities, notably of the effective diffusion coefficient, and to the proper formulation of the equations governing the macroscopic scale. Furthermore, we assumed in

this study that the macroscopic water vapor gradient is equal to the macroscopic gradient of saturated vapor, driven by the macroscopic thermal gradient. This assumption has been regularly made in the snow science community (Yosida et al., 1955; Colbeck, 1993; Sokratov and Maeno, 2000; Pinzer et al., 2012), and is supported by the idea that the ice in the snowpack tends to impose water vapor saturation at the macroscopic scale. It however remains possible that the macroscopic water concentration deviates from saturation, notably if the deposition and sublimation kinetics is slow. A rigorous upscaling method yielding the equations governing macroscopic water concentration would therefore also help quantifying if such a situation of non-saturation at the macroscopic scale is likely to occurs in real snowpacks, and indicate how the macroscopic vapor flux should be computed in such a case.

Finally, the fact that there is no macroscopic enhancement of the water vapor flux in snow suggests that most of the mass flux observed in subarctic and Arctic snow, and which would necessitate effective diffusion coefficients several times higher than that of free air to be explained solely by diffusion (e.g. Sturm and Benson, 1997; Domine et al., 2016, 2018), could rather be due to convection. The importance of convective mass transport in subarctic snowpacks has notably been pointed out by Trabant and Benson (1972) and Sturm and Johnson (1991), and thus appears as a good candidate to explain the high vapor movement in subarctic snowpacks. Currently, detailed snow physics models do not include the mechanism of convective mass transport (Lehning et al., 2002; Vionnet et al., 2012) and assume all mass transport to result from diffusion, sometimes using a diffusion coefficient larger than that in free air (e.g. Jafari et al., 2020). Further modeling efforts to include convective mass transport in detailed snow models could enhance their ability to model snowpack evolution.

## 7   Conclusions

This work investigated the macroscopic vapor fluxes that arise in snowpacks due to large scale vapor gradients. We first considered the seminal work of Yosida et al. (1955) and their formulation of the hand-to-hand delivery mechanism, which was meant to explain the large vapor flux they measured. We argue that it is reasonable to assume that the concentration of the thermal gradient in the pore space would lead to strong vapor gradients between ice grains, and drive the sublimation of water molecules from some grains and subsequent deposition on others. Yet, we disagree with the proposed idea that the process where one water molecule deposits on one side of an ice grain while an other molecule sublimates on the other side is equivalent to a situation where the depositing molecule skipped the ice, virtually increasing the vapor flux.

We demonstrated that the specific internal boundary conditions governing the sublimation and deposition of water molecules have a significant impact on the macroscopic vapor flux. In particular, we showed that in the case of infinitely fast kinetics the macroscopic flux is enhanced compared to the slow kinetics case, but still cannot exceed the vapor flux that would happen in free air under an equivalent vapor gradient. This demonstration is confirmed by numerical simulations on both idealized and measured snow microstructures. The discrepancies with previous studies that report vapor fluxes greater than the free air case originate from erroneous computations of how the macroscopic flux was obtained from the microscopic vapor fluxes at the pore scale. We argue that the method used in this article, i.e. volume averaging over an entire microstructure including the ice, is the only one consistent with the actual nature of the macroscopic water vapor flux.

The numerical simulations also indicate that the infinitely fast kinetics and inert ice surface cases respectively are the upper and lower limits for the vapor flux in snow. The use of more complex laws describing the sublimation and deposition of water molecules at the ice surface leads to flux values in between both previously mentioned cases. Moreover, the use of a non-constant attachment coefficient breaks the proportionality between the macroscopic vapor flux and the vapor gradient. In that case, it is no longer possible to define a constant and intrinsic effective diffusion coefficient, proportionally relating the macroscopic vapor flux to the macroscopic concentration gradient, independently of the magnitude of applied vapor concentration gradient.

*Code availability.* The codes used for the simulations were developed with python3 and ElmerFEM. They will be provided upon request to the corresponding author.

**Appendix A: Demonstration that the macroscopic vapor flux is maximal under infinitely fast kinetics**

The aim of this appendix is to demonstrate that the macroscopic vapor flux is maximal in the case of infinitely fast kinetics. For this we start by applying the spatial averaging theorem (Whitaker, 1999) to the vapor concentration in the pores $c$

$$\langle \nabla c \rangle = \nabla \langle c \rangle + \frac{1}{V} \int_{\Gamma} c \mathbf{n} \, \mathrm{d}S \tag{A1}$$

where $\langle \bullet \rangle$ is an operator defined as $\langle \bullet \rangle = \frac{1}{V} \int_{V_a} \bullet \, \mathrm{d}V$, and the concentration $c$ in the surface integral is the vapor concentration at the ice/pore interface. Multiplying by $D_0$, and using the notation introduced in this article for the macroscopic vapor flux $\mathbf{F}$, we have

$$\mathbf{F} = -D_0 \nabla \langle c \rangle - \frac{D_0}{V} \int_{\Gamma} c \mathbf{n} \, \mathrm{d}S \tag{A2}$$

Moreover, using the Hertz-Knudsen equation we have that the concentration at the interface is

$$c = c_{\mathrm{sat}} - \frac{D_0}{\alpha v_{\mathrm{kin}}} \nabla c \cdot \mathbf{n} \tag{A3}$$

Equation A2 can thus be written as

$$\mathbf{F} = -D_0 \nabla \langle c \rangle - \frac{D_0}{V} \int_{\Gamma} c_{\mathrm{sat}} \mathbf{n} \, \mathrm{d}S + \frac{D_0^2}{V \alpha v_{\mathrm{kin}}} \int_{\Gamma} (\nabla c \cdot \mathbf{n}) \mathbf{n} \, \mathrm{d}S \tag{A4}$$

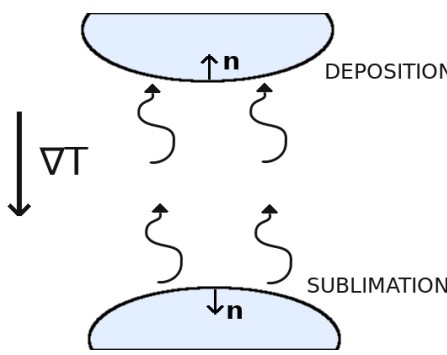

**Figure A1.** Schematic showing the normal vector **n** of deposition and sublimation surfaces. Ice crystals are represented in blue, and the thermal and vapor gradients are assumed to point downward.

Applying the same spatial averaging theorem to the saturation concentration $c_{\mathrm{sat}}$, we have

$$\frac{1}{V}\int_{\Gamma} c_{\mathrm{sat}}\,\mathbf{n}\,\mathrm{d}S = \langle\nabla c_{\mathrm{sat}}\rangle - \nabla\langle c_{\mathrm{sat}}\rangle \tag{A5}$$

Injecting Equation A5 in Equation A4 thus yields

$$\mathbf{F} = -D_0\nabla\langle c\rangle - D_0\langle\nabla c_{\mathrm{sat}}\rangle + D_0\nabla\langle c_{\mathrm{sat}}\rangle + \frac{D_0^2}{V\alpha v_{\mathrm{kin}}}\int_{\Gamma}(\nabla c\cdot\mathbf{n})\,\mathbf{n}\,\mathrm{d}S \tag{A6}$$

As we assume that the macroscopic vapor concentration equals the macroscopic saturation concentration gradient (as in
Yosida et al. (1955); Colbeck (1993); Sokratov and Maeno (2000); Pinzer et al. (2012)), we have that $\nabla\langle c\rangle = \nabla\langle c_{\mathrm{sat}}\rangle$. Thus

$$\mathbf{F} = -D_0\langle\nabla c_{\mathrm{sat}}\rangle + \frac{D_0^2}{V\alpha v_{\mathrm{kin}}}\int_{\Gamma}(\nabla c\cdot\mathbf{n})\,\mathbf{n}\,\mathrm{d}S \tag{A7}$$

Let us now assume, without loss of generality, that the macroscopic vapor and thermal gradients are orientated downward. As seen in Figure A1, surfaces that are characterized by a normal vector pointing upward are deposition surfaces. The product $\nabla c\cdot\mathbf{n}$ is therefore negative, and $(\nabla c\cdot\mathbf{n})\,\mathbf{n}$ is a vector pointing downward. Similarly, surfaces that are characterized by a normal
vector pointing downward are sublimation surfaces. The product $\nabla c\cdot\mathbf{n}$ is thus positive, and the vector $(\nabla c\cdot\mathbf{n})\,\mathbf{n}$ is pointing downward. Therefore, for both type of surfaces $(\nabla c\cdot\mathbf{n})\mathbf{n}$ is pointing downward. The surface integral term in Equation A7 thus acts in opposition of $-\langle D_0\nabla c_{\mathrm{sat}}\rangle$, and tends to reduce the macroscopic vapor flux. We thus have the inequality

$$|\mathbf{F}| \leq |\langle D_0\nabla c_{\mathrm{sat}}\rangle| \tag{A8}$$

We will now show that this upper bound is reached in the infinitely fast kinetics case. Indeed, under the infinitely fast kinetics hypothesis the product $\alpha v_{\mathrm{kin}}$ can be treated as going to infinity. At the same time, the surface integral of Equation A7 remains bounded, as the concentration gradient in the vicinity of the interface does not diverge. The surface integral thus vanishes, and the norm of the vapor flux is given by

$$|\mathbf{F}| = |\langle D_0 \nabla c_{\mathrm{sat}} \rangle| \tag{A9}$$

that is to say that the upper bound of the macroscopic vapor flux is reached under infinitely fast kinetics. Moreover, note that we re-derived that in the infinitely fast kinetics case, the macroscopic vapor flux is given by the spatial average of the saturation vapor concentration in the pore space.

## Appendix B: Saturation of vapor in the infinitely fast surface kinetics case

In the case of infinitely fast surface kinetics, and assuming a linear relation between saturation concentration and temperature, the equations governing the vapor concentration are

$$\begin{cases} \mathrm{div}(-D_0 \nabla c) = 0 & (\Omega_a) \\ c = c_{\mathrm{sat}} = AT + B & (\Gamma) \end{cases} \tag{B1}$$

where $A$ and $B$ are two constants characterizing the linear relationship between temperature and vapor concentration, and $T$ is temperature of the ice surface. Thanks to the linearity of the divergence and gradient operators, and owing to the fact that $\nabla B = 0$, the equations can be reformulated to

$$\begin{cases} \mathrm{div}(\nabla \theta) = 0 & (\Omega_a) \\ \theta = T & (\Gamma) \end{cases} \tag{B2}$$

where $\theta = (c - B)/A$ and we have used the fact that $D_0$ is a non-zero constant to eliminate it from the first equation. Moreover let us recall that in the air temperature $T_a$ is a solution of the following Laplace equation

$$\begin{cases} \mathrm{div}(\nabla T_a) = 0 & (\Omega_a) \\ T_a = T & (\Gamma) \end{cases} \tag{B3}$$

Systems of Equations B2 and B3 are identical, and since the solution of such a boundary value problem is unique it follows that $T_a = \theta = (c - B)/A$ over the entire pore space. It thus follows that $c = AT_a + B = c_{\mathrm{sat}}(T_a)$ in the pores.

**Appendix C: Vapor flux in the Hansen and Folsien, 2015 thermal conductivity**

Hansen and Foslien (2015) proposed that the heat flux $q_\mathrm{s}$ through a snow sample under a macroscopic thermal gradient $\nabla T$ be expressed as

$$q_\mathrm{s} = (1-\phi)q_\mathrm{tub} + \phi q_\mathrm{lam} \tag{C1}$$

where $q_\mathrm{tub}$ and $q_\mathrm{lam}$ are the heat fluxes through idealized snow structures corresponding respectively to a tubular structure and a lamellae structure, submitted to the same macroscopic thermal gradient $\nabla T$, and $\phi$ is the porosity of the snow sample (not to be confused with the ice volume fraction). Concerning the tubular microstructure, one has

$$q_\mathrm{tub} = (1-\phi)k_\mathrm{i} + \phi(k_\mathrm{a} + LD_0\frac{\mathrm{d}c_\mathrm{sat}}{\mathrm{d}T})\nabla T \tag{C2}$$

where $k_\mathrm{i}$ and $k_\mathrm{a}$ are the thermal conductivities of ice and air, and $L$ is the latent heat of sublimation of ice. The contribution of the vapor flux is $\phi LD_0\frac{\mathrm{d}c_\mathrm{sat}}{\mathrm{d}T}\nabla T$, and the vapor flux in the tubular microstructure is $\phi D_0\frac{\mathrm{d}c_\mathrm{sat}}{\mathrm{d}T}\nabla T = \phi D_0 \nabla C$.

Similarly one has concerning the lamellae microstructure

$$q_\mathrm{lam} = \frac{k_\mathrm{i}(k_\mathrm{a} + LD_0\frac{\mathrm{d}c_\mathrm{sat}}{\mathrm{d}T})}{(1-\phi)(k_\mathrm{a} + LD_0\frac{\mathrm{d}c_\mathrm{sat}}{\mathrm{d}T}) + \phi k_\mathrm{i}}\nabla T \tag{C3}$$

In their article, Hansen and Foslien (2015) then rewrite $q_\mathrm{lam}$ under the form

$$q_\mathrm{lam} = \frac{k_\mathrm{i}k_\mathrm{a}}{(1-\phi)(k_\mathrm{a} + LD_0\frac{\mathrm{d}c_\mathrm{sat}}{\mathrm{d}T}) + \phi k_\mathrm{i}}\nabla T + L\frac{\mathrm{d}c_\mathrm{sat}}{\mathrm{d}T}\frac{k_\mathrm{i}D_0}{(1-\phi)(k_\mathrm{a} + LD_0\frac{\mathrm{d}c_\mathrm{sat}}{\mathrm{d}T}) + \phi k_\mathrm{i}}\nabla T \tag{C4}$$

and identify the second term with the latent heat flux. We however argue that Equation C4 is only one way among many to rewrite $q_\mathrm{lam}$ under the form $A\nabla T + L\frac{\mathrm{d}c_\mathrm{sat}}{\mathrm{d}T}B\nabla T$, and thus that the identification of the latent heat flux with the second term of the decomposition is arbitrary.

To derive the latent heat flux, we first start from Equations 79 and 80 of Hansen and Foslien (2015) and compute the thermal
gradients in the ice and in the air, respectively, as

$$\nabla T_\mathrm{i} = \frac{q_\mathrm{lam}}{k_\mathrm{i}} = \frac{k_\mathrm{a} + LD_0\frac{\mathrm{d}c_\mathrm{sat}}{\mathrm{d}T}}{(1-\phi)(k_\mathrm{a} + LD_0\frac{\mathrm{d}c_\mathrm{sat}}{\mathrm{d}T}) + \phi k_\mathrm{i}}\nabla T \tag{C5}$$

and

$$\nabla T_\mathrm{a} = \frac{q_\mathrm{lam}}{(k_\mathrm{a} + LD_0\frac{\mathrm{d}c_\mathrm{sat}}{\mathrm{d}T})} = \frac{k_\mathrm{i}}{(1-\phi)(k_\mathrm{a} + LD_0\frac{\mathrm{d}c_\mathrm{sat}}{\mathrm{d}T}) + \phi k_\mathrm{i}}\nabla T \tag{C6}$$

The heat flux $q^{cond}$ through the sole process of conduction is thus given by

$$q^{cond} = (1-\phi)k_i \nabla T_i + \phi k_a \nabla T_a = \frac{k_i k_a + (1-\phi)k_i L D_0 \frac{dc_{sat}}{dT}}{(1-\phi)(k_a + L D_0 \frac{dc_{sat}}{dT}) + \phi k_i} \nabla T \tag{C7}$$

Meaning that the latent heat flux, which is the remaining contribution to $q_{lam}$, is $q_{lam} - q^{cond} = \frac{\phi k_i L D_0 \frac{dc_{sat}}{dT}}{(1-\phi)(k_a + L D_0 \frac{dc_{sat}}{dT}) + \phi k_i} \nabla T$, and that the vapor flux is $\frac{\phi k_i D_0}{(1-\phi)(k_a + L D_0 \frac{dc_{sat}}{dT}) + \phi k_i} \nabla C$. Note that the $\phi$ term in the numerator is not present in the original Hansen and Foslien (2015) demonstration, leading to an overestimation of the vapor flux.

Finally, the total vapor flux in the Hansen and Foslien (2015) model is computed as the weighted average of the tubular and lamallae vapor fluxes

$$F = \Big[ \phi \frac{\phi k_i D_0}{(1-\phi)(k_a + L D_0 \frac{dc_{sat}}{dT}) + \phi k_i} + (1-\phi)\phi D_0 \Big] \nabla C \tag{C8}$$

and the expression in square bracket is therefore the effective vapor diffusion coefficient, that one can show to be less than $D_0$.

## Appendix D: Physical constants

The physical constants used in this article are listed in Table D1, with their units, numerical values, and references.

**Table D1.** Physical constants used in the article

| Symbol | Signification | Value | Reference |
|--------|---------------|-------|-----------|
| $D_0$ | Diffusion coefficient of water vapor in the air | $2 \times 10^{-5} \, \mathrm{m^2 \, s^{-1}}$ | Calonne et al. (2014) |
| $P_0$ | Saturation pressure of water vapor over ice at $273 \, \mathrm{K}$ | $611 \, \mathrm{Pa}$ | Lide (2006) |
| $\Delta H_s$ | Latent heat of sublimation of ice | $28 \times 10^5 \, \mathrm{J \, kg^{-1}}$ | Lide (2006) |
| $k_i$ | Thermal conductivity of ice | $2.34 \, \mathrm{W \, K^{-1} \, m^{-1}}$ | Riche and Schneebeli (2013) |
| $k_a$ | Thermal conductivity of air | $0.024 \, \mathrm{W \, K^{-1} \, m^{-1}}$ | Riche and Schneebeli (2013) |

*Author contributions.* FD designed research with inputs from PH and KF. FD obtained funding. KF performed research and wrote the paper with inputs from FD and PH.

*Competing interests.* The authors declare having no competing interests.

*Acknowledgements.* This work contributes to the APT project (Acceleration of Permafrost Thaw), funded by the Climate Initiative program of the BNP-Paribas Foundation. We thank Jacques Roulle for his help during the tomography imaging and cold-room work. Neige Calonne and Marie Dumont provided valuable inputs on the subject. Henning Löwe kindly indicated the usage of the spatial averaging theorem. We thank Kevin Hammonds and Quirine Krol for their helpful review of the manuscript, and Jürg Schweizer for editing the article.

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
