# Peer review of "Macroscopic water vapor diffusion is not enhanced in snow"

_The Cryosphere, 2020_

## Referee Comment (RC1) · Kevin Hammonds (Referee) · 19 Sep 2020

Kevin Hammonds (Referee)

kevin.hammonds@montana.edu

To the Authors and the EiC,

I have read and reviewed the article, "Macroscopic water vapor diffusion is not enhanced in snow", submitted for publication in The Cryosphere by K Fourteau, F Domine, and P Hagenmuller. In this study, the authors investigate via theoretical considerations of diffusion and attachment kinetics combined with numerical simulations, whether or not water vapor diffusion in snow is enhanced on the macro-scale, when compared to water vapor diffusion in air. The authors frame the historical context of their study by providing a detailed overview of previous work on the topic, beginning with Yosida et al. (1955), that have led to the commonly held belief that water vapor

diffusion is enhanced in snow at the macro-scale, due to "hand-to-hand" mechanisms of water vapor transport occurring at the micro-scale. The authors challenge this explanation by i) deriving a theoretical model from first principles that accounts for both attachment kinetics and diffusion within an inert or kinetically active porous medium and ii) performing numerical simulations on both idealized snow microstructures and actual snow microstructures obtained from micro-CT. With this approach, the authors show that although the diffusion of water vapor in the pore-space between ice grains can be greater than that of water vapor in air, this effect is more than countered by the overall tortuous nature of the ice matrix, such that diffusion alone cannot account for the sometimes large water vapor flux observed in snow at the macro-scale, even with infinitely fast and/or non-linear kinetics.

Substantively, my only comments relate to potential areas where the paper could be improved by further explaining or providing evidence for why certain terms/processes were either neglected or only speculated upon in their study. For instance, to explain field observations of a larger than expected water vapor flux in snow, the authors suggest that convection may also be occurring at the macro-scale, but present no evidence in support of this speculation. Furthermore, the effects of ice grain curvature on the overall vapor flux are neglected, also without explicit evidence or discussion. Last, the effects of latent heat are neglected as well, again without sufficient evidence. While I would generally agree that the effects of ice grain curvature and latent heating are not primary drivers of water vapor diffusion in snow when a strong temperature gradient is also present, it is my opinion that the manuscript could be strengthened and the broader impacts increased if further explanation were provided with regards to these and other topics.

Overall, I found the article to be convincing, well-written, and worthy of publication in The Cryosphere. Furthermore, it should be noted that the topic at hand has been one of debate, and it is not without due deliberation that I submit my review. I congratulate the authors on the merits of their work and acknowledge the respectful tone with which

they address the historical significance of their findings. Additional comments below.

Recommendation: Minor Revisions

Best regards,

Kevin

Kevin Hammonds, Ph.D. Subzero Research Laboratory, Dept. of Civil Engineering Montana State University, Bozeman, MT, USA https://www.montana.edu/subzero/

General Comments:

1) Where Colbeck 1983 is cited for neglecting any contribution to the vapor flux from the local curvature of ice (lines 155-157), I think an opportunity is missed here to acknowledge and discuss some of the more recent work on the topic (Krol & Lowe 2016, 2018) that have also investigated local ice crystal growth rates as a function of curvature from micro-CT. Similarly, these authors also develop a treatment of vapor diffusion in snow that accounts for the full ice matrix at the pore scale, that they claim can be scaled to larger length scales, such that their work seems equally relevant in that regard as well.

2) Although it is mentioned in the Discussion appropriately and with references, I would recommend removing any mention of convection as a possible process by which water vapor transport is occurring in snow from the Abstract, as there is no evidence provided in support of this statement explicitly from this study.

3) Could you further explain your reasoning for neglecting the latent heat flux in your treatment (lines 290-292)? In Hammonds and Baker 2016 (Figure 7), it was estimated that the latent heat flux from deposition may have accounted for approximately 10% of the increase of the local temperature gradient (these calculations were based on Riche and Schneebeli 2013). Furthermore, whether or not the latent heat is expected to be absorbed into the ice matrix or released into the surrounding air upon phase change would also be of relevance for increasing or decreasing the local temperature gradient. Experimental SEM observations from Hammonds et al. 2015 also address this point,

[Figure]

Last, Libbrecht and Rickerby 2013 (Section 2.3) also discuss the likelihood for latent heat flux to be released or absorbed as a function of ice crystal size.

4) Figure 1: This illustration and explanation (lines 87-90) could be improved by also showing the case of the "ice phase" (here, phase is used correctly) that is just above or just below the two cans, such that if one calculated the net mass flux for all three cases across this boundary, a zero net mass flux is observed.

5) Is it possible to be more specific about the separation of scales (line 105)? Is Lmicro « Lmacro sufficient? Molecular attachment, for instance, occurs on a length scale even smaller than Lmicro. Please comment.

6) What is different about ice crystal growth in a snowpack (Line 318 – 319)? For faceted ice crystal growth, such is presented in this study, the molecular attachment considerations presented in Libbrecht and Rickerby 2013 seem sufficient. Furthermore, theory would dictate that attachment from the vapor phase would be preferred on the primary prism face (Brumberg et al 2017). Recommend deleting "and might not properly apply to ice in snowpacks".

7) Please include an additional item in the Appendix that presents the numerical values used for each constant given in each equation, with units and a citation for each.

8) Throughout the manuscript, the word "phase" is used to represent what I think is meant to be "space", as in "pore phase" or "air phase". I would recommend using the word "space" instead of "phase", and reserving "phase" only for referencing the thermodynamic state (i.e. solid, liquid, or gas).

9) Throughout the manuscript, the words "inferior" or "superior" are used to represent "less than" or "more than", in terms of comparing two quantities. As the terms "inferior" and "superior" are often used in English with connotations of mediocrity or greatness, respectively, would recommend just using "less than" or "greater than" throughout the manuscript.

Specific Comments:

Line 52: The cans filled with snow were "weighed" (not "weighted")

Line 109: I am not sure what is meant by "solicitations" in this context.

Line 111: Add a "t" . . .but not so large that it spans several. . .

Line 118: switch order of "time unit" to "unit time"

Line 133: Can you provide a citation for the use of an Effective Diffusion Coefficient?

Line 146: Instead of "submitted to", perhaps "placed under" or "held under" would be more appropriate in this context.

Line 200: Instead of "1/phi factor", should be "a factor of 1/phi"

Line 255: To be more technically correct, replace "tomography scanning" with "X-ray computed microtomography (micro-CT)" or similar. (also in line 377)

Line 293: Here and throughout the article, could you provide references for the values chosen?

Line 335: "act as a blockage"

Line 336: "go around" not "goes around"

Line 382: Replace "zoom" with "focused view" or similar. . .

Table 1: Please change the label "Inf. Fast Kinetics" to "Deff", as this more accurately describes these quantities.

Line 437: Remove "?" from Domine citation

Line 438: "snowpacks" not "snowpack" in this context

References:

Brumberg, A., Hammonds, K., Baker, I., Backus, E. H., Bisson, P. J., Bonn, M., ... &

Shultz, M. J. (2017). Single-crystal Ih ice surfaces unveil connection between macroscopic and molecular structure. Proceedings of the National Academy of Sciences, 114(21), 5349-5354.

Krol, Q., & Löwe, H. (2016). Analysis of local ice crystal growth in snow. Journal of Glaciology, 62(232), 378-390.

Krol, Q., & Löwe, H. (2018). Upscaling ice crystal growth dynamics in snow: Rigorous modeling and comparison to 4D X-ray tomography data. Acta Materialia, 151, 478-487.

Hammonds, K., Lieb-Lappen, R., Baker, I., & Wang, X. (2015). Investigating the thermophysical properties of the ice–snow interface under a controlled temperature gradient: Part I: Experiments & Observations. Cold Regions Science and Technology, 120, 157-167.

Hammonds, K., & Baker, I. (2016). Investigating the thermophysical properties of the ice–snow interface under a controlled temperature gradient Part II: Analysis. Cold Regions Science and Technology, 125, 12-20.

Riche, F., & Schneebeli, M. (2013). Thermal conductivity of snow measured by three independent methods and anisotropy considerations. The Cryosphere, 7(1), 217.

Please also note the supplement to this comment:
https://tc.copernicus.org/preprints/tc-2020-183/tc-2020-183-RC1-supplement.pdf
* * *

---

## Referee Comment (RC2) · 22 Sep 2020

**Review: Macroscopic water vapor diffusion is not enhanced in snow**

**General comments**

This paper addresses the long standing controversy on effective water vapor diffusion in snow. The paper is a welcome contribution to the topic, and include numerical simulations showing that the effective diffusion is not enhanced. Although the paper is missing a rigourous mathematical derivation of the statements on the definition of the effective vapor diffusion coefficient, its bounds and upscaling approach, it makes up for it by the valuable assessment of the dependency of the effective water vapor transport in snow on the accommodation coefficient $\alpha$. The authors go at great length into the history and details of the problem and the different angles previous studies took, and along the way narrow down how the effective diffusion coefficient should be defined. The paper describes a rich number of simulations to solve the coupled static heat and mass flux equations with Robin boundary conditions: introducing the Hertz-Knudsen-Langmuir equation and is the first of its kind within the snow microstructure community. This equation is primarily dependent on the accommodation/sticking/condensation/sublimation/... coefficient. This equation is introducing a natural way to continuously move from an inert media with no crystal growth, to locally enhanced diffusion driven by local sinks ans sources throughout the media. This setup enables to compute the effective diffusion coefficient defined by total volume averaged mass flux divided by macroscopic water vapor concentration gradient.

Overall the paper is well written, including a clear motivation, strong methods, and reasonable conclusions, and I congratulate the authors with this work. In principle it can be published with minor revisions, since all the computations and simulations are, to an acceptable degree physically sound, and the results will prove valuable to the snow physics community.

The comments that follow are in general a matter of taste and representation. That said, I think the paper deserves a more classical setup (introduction, theoretical background, methods, results...). Especially a clear description that starts with a formal definition of the involved equations. An unambiguous mathematical upscaling method (volumetric averaging) is desired and would help the reader to be convinced by the conclusions of the paper. Given the fact that effective diffusion at the microscale is difficult to measure experimentally, such a study that includes simulations that resolve the water vapor concentration at the microscale deserves a central approach. In my opinion this paper should be primarily centered around the simulations and the influence of finite kinetics to the overall water vapor transport and secondary on how it relates to previous (experimental) studies. The latter can be discussed at length in the discussion section. The title could also reflect the importance of the influence of finite kinetics to the effective diffusion coefficient. In general I would encourage the authors to refocus the manuscript in the formerly described manner.

Comments that refer to restructuring of the whole manuscript are optional, others are considered to be essential (bold-faced).

General comments:

1 The chosen upscaling method of 'volumetric averaging' over 'cross-section averaging' (l.120 -l.132) is based on the argument that microscopic scale variations are not accessible by area averaging. To my understanding this is an issue related to the Representative Element Volume (REV). Snow microstructures are measured with $\mu$CT, large enough such that the volume is representative and homogeneous

in a volumetric manner. If cross-sections are used this might not be satisfied anymore as rightly addressed by Pinzer et al. [2012] and volumetric averaging can be chosen. It is therefore not the intrinsically preferred method, but one that is dictated by the specific microstructure.

2 The chosen upscaling method is important especially if we couple the effective diffusion to the macroscopic mass and heat transport Calonne et al. [2014]. This study should explicitly relate its results to this study, and how these equations should be adapted.

3 The accommodation coefficient, including its name, should be introduced in the introduction including experimental observations such as Libbrecht [2005], Harrington et al. [2019] and possibly other studies. The choice of values for the simulations should be linked and/or motivated by deficiencies of these studies.

4 Although symbols in equations are generally well described and it is clear from the context what they mean, it might be helpful to the reader to introduce systematic notation to distinguish between upscaled quantities and local quantities, e.g.

$$F = \frac{1}{V} \int_V f \, dx^3, \tag{1}$$

in other words, how are $F$ and $C$ related to their microscopic quantities?

5 In case of volume averaging, gradients of microscopic fluxes are influenced by sources and sinks at internal ice-air iterfaces Whitaker [1998], Krol and Löwe [2018], i.e.

$$\langle \boldsymbol{\nabla} f \rangle = \boldsymbol{\nabla} \langle f \rangle + \int_\Gamma f d\boldsymbol{n}. \tag{2}$$

In case of the idealized spheres the second term vanishes because of symmetry, but for your snow samples it might not be the case, and should be shown, either by estimating the order of magnitude of the gradient of your sources and sinks, or by analysis of the simulations that this term is rightfully neglected. Here it matters how the macroscopic quantities are related to their microscopic counterparts. Note that in your simulations you average over both phases, vapor and ice, but you neglect the sinks and sources. I believe with these microstructures it is probably alright, but it should be estimated/shown that you can do so.

**Specific comments**

l.10 Naming of the coefficient $\alpha$. This coefficient is often related to the phase-change it represents i.e. deposition, sublimation, or sticking parameter.

l.11 There is no evidence or discussion in the paper that suggests that convection is one of the candidates responsible for the experimentally observed mass deficiency.

l.46 Suggestion to shorten this paragraph and move to the discussion. The notion of hand-to-hand diffusion should be discarded on the fact this is simply no physical transport of water molecules.

l.120-132 Please be very specific about your methods of upscaling. See general comments 1, 2, and 5.

l.133 Here I would expect a mathematical definition, including upscaling methods, see comment 5.

l.137 Semantic comment: What does 'ideally' mean in this context? Maybe include that intrinsic, in this context, means that $D_{\text{eff}}$ is independent of the external temperature gradient. When $D_{\text{eff}}$ is dependent on the external gradient, one could say that the response of the material is non-linear. Does this break the definition of the effective diffusion coefficient, meaning the coefficient that quantifies the vapor flux as a linear response to an applied concentration gradient?

**l.150** In this paragraph I suspect at least an expression for the the macroscopic vapor flux as suggested by the title.

**l.165 and l.308** . How infinite can $\alpha v_{\text{kin}}$ be? $v_{\text{kin}}$ is finite $\sim 10^2$, and $0 < \alpha < 1$. In principle it should be compared to the actual interface velocity $v_n$ in the Robin b.c. as stated in Kaempfer and Plapp [2009]. A discussion on $\alpha$ and its values would be appreciated Libbrecht [2005], Saito [1996], Legagneux and Dominé [2005].

l.263 This paragraph includes an important realization, how does it relate to the expression for the macroscopic heat transport provided by Calonne et al. [2014]. This could be treated in the discussion.

**l.282** Some more details on the technicalities of the simulation should be provided, are $T$ and $c$ computed simultaneously? or is $c$ computed given $T$? How is it parallelized, and how long does it take? What are the meshing requirements, how many points etc.

**l.367** For the non-linear kinetics results it might be useful to state the surface averaged simulated $\alpha$ and its variance.

**l.367** How sensitive is your result to the value $\sigma_0$? Since it might differ for different crystallographic surfaces.

**Fig.4** I suggest to split this plot into two figures. One for linear simulations $D_{\text{eff}}^{\text{norm}}$ vs $\alpha$ and the other for non-linear dynamics $D_{\text{eff}}^{\text{norm}}$ vs $\nabla T$ including colorbar for surface averaged $\alpha$. This suggestion is given to observe the type of transition between purely tortuous diffusion and phase transition enhanced diffusion. The data on the non-linear dynamics seems to rapidly depart from the tortuous diffusion case: is there a reason for this? We would expect also here a smoother transition between the two limiting cases, such as in Fig.6. The results for small temperature gradients puzzle me. A discussion on the results in this regime might be helpful.

**l.402** Moreover? Is there a reason not to compute the non-linear cases? In my opinion it is interesting and worth it to quantify the different non-linear responses of the 6 different snow types.

**Fig.7 and Table 1** The Figure and Table have approximately the same information. Consider plotting again $D_{\text{eff}}^{\text{norm}}$ vs $\alpha$ and colorbar on density. Alternative, plot $D_{\text{eff}}^{\text{norm}}/\phi$ and discuss the remaining influence of SSA. If SSA is presented in either a table or a plot, then a note in the discussion on its influence is desirable.

l.440 A list of the general causes to why vapor flux was considered to be enhanced in the past is expected in the discussion.

**l.440** A reasonable explanation for why convection could be the cause of the experimentally observed mass deficit could go here.

**l.448** 'Disagree', is an understatement. You show with numerical simulation that this concept is ill-defined. Suggestion: We show with numerical simulations that increased vapor flux by the hand-to-hand mechanism is not present.

**l.458** Avoid 'intuitive'. Suggestion: consistent with actual water vapor transport.

**Appendix B, l.492** incorrect use of 'inferior', use 'less than'.

**Technical corrections** Overall technical comments,

**l.59** The use of pore phase, throughout the manuscript is incorrect. Please use pore space, or vapor/gas phase. Also air phase is not commonly used.

**Overall** The use of colons is not consistent, e.g. before equations introduced by 'given by' it is not very common to use them. Use of colons is generally restricted to lists or 'may' be used between independent clauses when the second sentence explains, illustrates, paraphrases, or expands on the first sentence. Equations are part of sentences and therefore colons should not appear more often before an equation than in other parts of your text.

**l.296** Outer brackets in the exponent should be larger, (use \left( and \right) commands).

**l.304** Condensation is reserved for the gas-liquid phase-transition, use deposition (or desublimation) also at other places throughout the manuscript.

**l.336** Goes → go.

**l.437** ? citation missing.

**l.454** Similar → Equivalent.

**References**

N. Calonne, C. Geindreau, and F. Flin. Macroscopic modeling for heat and water vapor transfer in dry snow by homogenization. *J. Phys. Chem. B*, 118(47):13393–13403, 2014. doi: 10.1021/jp5052535.

J. Y. Harrington, A. Moyle, L. E. Hanson, and H. Morrison. On Calculating Deposition Coefficients and Aspect-Ratio Evolution in Approximate Models of Ice Crystal Vapor Growth. *J. Atmos. Sci.*, 76(6):1609–1625, June 2019. ISSN 0022-4928, 1520-0469. doi: 10.1175/JAS-D-18-0319.1.

T. U. Kaempfer and M. Plapp. Phase-field modeling of dry snow metamorphism. *Phys. Rev. E*, 79(3, Part 1), 2009. ISSN 1539-3755. doi: 10.1103/PhysRevE.79.031502.

Q. Krol and H. Löwe. Upscaling ice crystal growth dynamics in snow: Rigorous modeling and comparison to 4D X-ray tomography data. *Acta Materialia*, 151:478–487, 2018. ISSN 1359-6454. doi: 10.1016/j.actamat.2018.03.010.

L. Legagneux and F. Dominé. A mean field model of the decrease of the specific surface area of dry snow during isothermal metamorphism. *J. Geophys. Res. Earth*, 110: F04011, 2005. doi: 10.1029/2004JF000181.

K. G. Libbrecht. The physics of snow crystals. *Rep. Prog. Phys.*, 14(4):599–895, 2005.

B. R. Pinzer, M. Schneebeli, and T. U. Kaempfer. Vapor flux and recrystallization during dry snow metamorphism under a steady temperature gradient as observed by time-lapse micro-tomography. *The Cryosphere*, 6(5):1141–1155, 2012. doi: 10.5194/tc-6-1141-2012.

Y. Saito. *Statistical Physics of Crystal Growth.* World Scientific, 1996.

S. Whitaker. *The Method of Volume Averaging.* Theory and Applications of Transport in Porous Media. Springer Netherlands, 1998. ISBN 978-0-7923-5486-4.

---

## Author Comment (AC1) · 21 Oct 2020

**TC-2020-183**

**RESPONSE TO KEVIN HAMMONDS**

We are thankful to Kevin Hammonds for the thorough review of our manuscript and its constructive comments.
We have copied his comments below in light blue, and provided our answers in black below them. Modifications to the text of the manuscript are proposed in green.

Note that we have discovered two numerical errors in the previous version of the manuscript that we we will correct in the new version:
- **L343**: The average gradient in the air is 79.00 K/m (and not 77.57 K/m as previously stated).
- **Table 1** and **Fig 7**: The effective diffusion coefficients for the Melt Forms sampled have been underestimated by 33%.

**GENERAL COMMENTS :**

1) Where Colbeck 1983 is cited for neglecting any contribution to the vapor flux from the local curvature of ice (lines 155-157), I think an opportunity is missed here to acknowledge and discuss some of the more recent work on the topic (Krol & Lowe 2016, 2018) that have also investigated local ice crystal growth rates as a function of curvature from micro-CT. Similarly, these authors also develop a treatment of vapor diffusion in snow that accounts for the full ice matrix at the pore scale, that they claim can be scaled to larger length scales, such that their work seems equally relevant in that regard as well.

The study of Krol and Loewe (2016) experimentally studied isothermal and thermal gradient metamorphism, and discuss whether a diffusion-limited hypothesis (roughly corresponding to a high $\alpha$ in our article) or a kinetics-limited hypothesis (low $\alpha$) is more consistent with their observations. Note that they do not study the macroscopic vapor flux but the interface velocities (which are related but not equivalent to one another). They observe that isothermal crystal growth is slightly better represented by the kinetics-limited case while gradient metamorphism is well represented by diffusion-limited kinetics and assuming water vapor saturation at the interface ice/pore. Adding the effect of curvature in their derivations improve the quality of their fit to their temperature gradient metamorphism data.

While we fully agree that curvature effects might play a role for the growth of ice crystals, we think that the impact on the macroscopic flux is negligible. Indeed, for curvature effects to drive a net macroscopic flux, a macroscopic gradient of curvature should be present in the snow layer, which is unlikely to be the case in homogeneous snow layers. We are currently not able to test this hypothesis, but future work could be to add a curvature term in the saturated water vapor concentration and see the impact of the macroscopic flux.

We will add details on why we neglect curvature effects, **L155**:
"In the presence of a large enough thermal gradient, the dependence of the saturation concentration to the local curvature of the ice surface can be neglected compared to its dependence on temperature (Colbeck, 1993). Under this condition, we can expect $c_{sat}$ to becomes a function of temperature only. Moreover, even if curvature effects were not negligible at the microscopic level it appears unlikely for them to results in a net macroscopic vapor flux. Indeed, in an homogeneous snow layer curvature differences are distributed isotropically within the microstructure, and thus do not result in a net movement of water vapor."

We will also refer to the study of Krol and Loewe (2016) and their experimental observations in the discussion, **L425**:

" […] we observe that $D_{eff}$ / (phi $D_0$) is almost always greater than unity, which supports the notion of fast rather than slow kinetics. This is consistent with the study of Krol and Löwe (2016) that report that fast kinetics is consistent with their microtomography-based observation of the temperature gradient metamorphism of a snow sample."

The study of Krol and Loewe (2018) propose equations for the geometrical evolution of the averaged SSA and curvatures (assuming that the interface velocity is known in the microstructure). They do not treat the vapor flux in the pores, and therefore we do not see how this study can be compared to our.

2) Although it is mentioned in the Discussion appropriately and with references, I would recommend removing any mention of convection as a possible process by which water vapor transport is occurring in snow from the Abstract, as there is no evidence provided in support of this statement explicitly from this study.

We agree and will remove the mention of convection in the abstract. We will reformulate the discussion on convection to **L437**:

"Indeed, the importance of convective mass transport in subarctic snowpacks has notably been pointed out by Trabant and Benson (1972) and Sturm and Johnson (1991), and thus appears as a good candidate to explain the high vapor movement in subarctic snowpacks."

3) Could you further explain your reasoning for neglecting the latent heat flux in your treatment (lines 290-292)? In Hammonds and Baker 2016 (Figure 7), it was estimated that the latent heat flux from deposition may have accounted for approximately 10% of the increase of the local temperature gradient (these calculations were based on Riche and Schneebeli 2013). Furthermore, whether or not the latent heat is expected to be absorbed into the ice matrix or released into the surrounding air upon phase change would also be of relevance for increasing or decreasing the local temperature gradient. Experimental SEM observations from Hammonds et al. 2015 also address this point. Last, Libbrecht and Rickerby 2013 (Section 2.3) also discuss the likelihood for latent heat flux to be released or absorbed as a function of ice crystal size.

Our understanding is that the work Hammonds and Baker 2016 quantifies the effect of latent heat on the macroscopic thermal gradient (the thermal gradient of snow at the layer scale), and not the temperature gradient within the pores themselves. On the other hand, the effective diffusion coefficient of vapor is rather governed by the ratio of thermal gradient in the pore space (governing the concentration gradient in the pores) over the macroscopic thermal gradient (governing the macroscopic concentration gradient).

At the pore scale, water vapor transports heat from the warm sublimation surfaces towards the cold deposition surfaces. In a sense, it acts as a heat transport mechanism in the pore space that occurs in parallel of heat conduction. Taking latent heat into account at the microscopic scale thus can be viewed as artificially increasing the heat conduction of air. This reduces the thermal contrast between the ice and pore spaces, and decreases the thermal gradient in the low-conducting pore space (all this working at constant macroscopic thermal gradient). Decreasing the thermal gradient in the pore space decreases the vapor concentration gradient in the pore space, resulting in a lower macroscopic vapor flux and a lower effective diffusion coefficient.

At the microscopic scale, the release and absorption of latent heat appears as a discontinuity of the heat conduction flux at the ice/pore interface (Calonne et al., 2014). From our understanding latent

heat thus warms/cools both the ice and the air in the vicinity of the interface. The degree to which the surrounding air and ice are impacted depends on the ability of the heat to be conducted away in the ice or in the air spaces though their thermal conductivities.

The discussion of Libbrecht and Rickbery (2013) and the "bread-loafing" observations of Hammonds and Baker (2016) focus on whether the deposition zones are heated enough to disrupt the anisotropic crystallization of water, and to influence the crystal habits. In our case we are only concerned with the vapor flux in the pores, and do not treat crystal habits. We therefore do not see how to integrate such observations in our treatment, apart from the fact that they are consistent with the notion of warming deposition surfaces and cooling sublimation surfaces.

To quantify this effect we have performed an additional FEM simulation. In the case of a fully vapor saturated pore space (corresponding to the infinitely fast kinetics in our case), Yosida et al. (1955) argue that latent heat effects at the microscopic scale can be treated by increasing the thermal conductivity of air by $dc_{sat} / dT * D_0 * L$, where L is the latent heat of sublimation of ice. We integrated this in a FEM simulation of the DF snow sample, and found that the effective diffusion coefficient dropped from 0.982 to 0.980.  The results of this double diffusion problem simulation, indicate that consistently with our understanding, latent heat effects reduce the macroscopic vapor flux at constant macroscopic thermal gradient and thus diminishes the effective diffusion coefficient of vapor. Moreover, the overall impact appears to be quantitatively small.

We will add to the text **L291**:
"At the microscopic level, adding latent heat effects would act as an additional mechanism transporting heat from the warm sublimating surfaces towards the cold deposition surfaces. It would cool the sublimation surfaces and warms the deposition surfaces, decreasing the thermal gradient in the pore space. Therefore, taking latent heat effects into account would not increase the effective vapor diffusion coefficient."

4) Figure 1: This illustration and explanation (lines 87-90) could be improved by also showing the case of the "ice phase" (here, phase is used correctly) that is just above or just below the two cans, such that if one calculated the net mass flux for all three cases across this boundary, a zero net mass flux is observed.

We are not sure to fully understand the comment of the reviewer. The cases of ice grains just below or just above the boundary can be treated exactly as ice grains far away from it: the ice appearing or disappearing in the volume control correspond to water molecules already present in the volume under the gaseous form. Therefore, no mass transport is associated with the deposition and sublimation of water molecules.

We think that showing ice grains just below and above the boundary would give the wrong impression that the gain of ice of the grain just above the boundary should be understood as compensated by the loss of mass of the grain just below it.

5) Is it possible to be more specific about the separation of scales (line 105)? Is Lmicro « Lmacro sufficient? Molecular attachment, for instance, occurs on a length scale even smaller than Lmicro. Please comment.

Our goal in this article is to see what can be said about the macroscopic vapor flux starting from equations directly at the pore scale. It is true that the microscopic equations are themselves "homogenized" from the atomic scale, for instance by encompassing all the processes of water molecules attachment onto ice within a single α parameter.

However, as we start from the pore scale, it is not necessary for us to explicitly treat the atomic scale (even though a good understanding of the atomic scale physics is helpful to develop a good comprehension of the pore scale physics). Similarly to Calonne et al. (2014), the validity of the pore scale equations are taken as a given in the article.

6) What is different about ice crystal growth in a snowpack (Line 318 – 319)? For faceted ice crystal growth, such is presented in this study, the molecular attachment considerations presented in Libbrecht and Rickerby 2013 seem sufficient. Furthermore, theory would dictate that attachment from the vapor phase would be preferred on the primary prism face (Brumberg et al 2017). Recommend deleting "and might not properly apply to ice in snowpacks".

The measurements of Libbrecht and Rickerby 2013 are based on the growth of faceted crystals, which only present a limited number of ideal surface types. In real snowpacks however, the variety of crystal surface types is much greater, with non-flat and vicinal surfaces, such as sublimation surfaces that tend to be rounded. Such surfaces behave differently from facets as deposition is facilitated on them, due to the greater density of steps. Moreover, we assume in the article that deposition and sublimation physics are symmetric, which might however not be the case. Therefore, we cannot guarantee that the simple non-linear law used in this article applies to the entirety of a snow sample. The quantitative use of non-linear kinetics would first require to properly characterize the crystallographic state of the different surfaces within a snow sample, something we cannot do at the moment.

We will add more justifications on why we think that while non-linear surface kinetics is a promising topic for snowpack physics, our results in this study should be viewed as purely qualitative and should not be over interpreted. For this we will add **L355**:
"[…] and might not properly apply for the entirety of ice surfaces in snowpacks. Indeed, this law has been derived using deposition measurement, and might not apply for sublimation surfaces (Beckmann and Lacmann, 1982). Moreover, the presence of vicinal surfaces in snowpacks, where the proposed law does not apply, is likely (Legagneux and Domine, 2005). Therefore, the point of using such a law is to qualitatively study the potential impact of a dependence of $\alpha$ to the local vapor saturation, rather than to produce quantitative results."
We will also justify our choice not to use the non-linear kinetics law on all of our snow sample **L402**:
"We also did not compute $D_{eff}$ with non-linear surface kinetics (i.e. when $\alpha$ is not constant), as we are not confident in the validity of the chosen non-linear law for snow modeling."

7) Please include an additional item in the Appendix that presents the numerical values used for each constant given in each equation, with units and a citation for each.

We will add a table with the numerical values of constants and appropriate reference in the Appendix, and refer to it in the main part of the manuscript.

8) Throughout the manuscript, the word "phase" is used to represent what I think is meant to be "space", as in "pore phase" or "air phase". I would recommend using the word "space" instead of "phase", and reserving "phase" only for referencing the thermodynamic state (i.e. solid, liquid, or gas).

We will remove the word phase throughout the manuscript when it is not appropriate.

9) Throughout the manuscript, the words "inferior" or "superior" are used to represent

"less than" or "more than", in terms of comparing two quantities. As the terms "inferior" and "superior" are often used in English with connotations of mediocrity or greatness, respectively, would recommend just using "less than" or "greater than" throughout the manuscript.

We will change the "inferior" and "superior" to "less than" and "greater than" throughout the manuscript.

**SPECIFIC COMMENTS:**

Line 52: The cans filled with snow were "weighed" (not "weighted")
We will correct the spelling.

Line 109: I am not sure what is meant by "solicitations" in this context.
Here by solicitations we refer to the external physical conditions imposed to the sample, might it be the thermal and vapor gradients or imposed thermal and vapor fluxes. The employed term is usually "excitation", as in Calonne et al. (2014) for instance.

We will rephrase **L109** to:
"[…] $L_{macro}$ is the length-scale characterizing variations of the snowpack or of the external forcing applied at the macroscopic scale, for instance the change between different snow layers or changes in thermal and vapor gradients"

Line 111: Add a "t" . . .but not so large that it spans several. . .
We will correct the typo.

Line 118: switch order of "time unit" to "unit time"
We will rephrase accordingly.

Line 133: Can you provide a citation for the use of an Effective Diffusion Coefficient?
We will provide references, within and outside of the snow science community, namely Colbeck (1993), Calonne et al., (2014), and Bourbatache et al. (2020).

Line 146: Instead of "submitted to", perhaps "placed under" or "held under" would be more appropriate in this context.
We will rephrase to:
"Let us consider a volume of snow (Figure 2 a), subjected to vertical macroscopic temperature and vapor gradients at its boundaries."

Line 200: Instead of "1/phi factor", should be "a factor of 1/phi"
We will rephrase this sentence accordingly, as well as in the sentence **L253**.

Line 255: To be more technically correct, replace "tomography scanning" with "X-ray computed microtomography (micro-CT)" or similar. (also in line 377)
We will use the term of "X-ray computed microtomography" in the article.

Line 293: Here and throughout the article, could you provide references for the values chosen?
We will provide the references within a new Appendix presenting all the physical constants used.

Line 335: "act as a blockage"
We will rephrase accordingly.

Line 336: "go around" not "goes around"
We will correct the typo.

Line 382: Replace "zoom" with "focused view" or similar. . .
We will rephrase to:
"A close-up view showing the vapor stream lines [...]"

Table 1: Please change the label "Inf. Fast Kinetics" to "Deff", as this more accurately describes these quantities.
The label Inf. Fast Kinetics refers to the value of $\alpha$ used for the simulations (in this case $\alpha \rightarrow$ infinity). We will redo the table to clearly indicate that all numerical values in the right part of the table are all $D_{eff}$ values.

Line 437: Remove "?" from Domine citation
There should be a citation of Sturm and Benson (1997) instead of the "?", but we did not realize that the citation link was broken. This will be fixed in the new version.

Line 438: "snowpacks" not "snowpack" in this context
We will rephrase accordingly.

---

## Author Comment (AC2) · 21 Oct 2020

TC-2020-183

**RESPONSE TO QUIRINE KROL**

We are thankful to Quirine Krol for her detailed and constructive review of our manuscript.

We in general agree with the insightful remarks of the reviewer and will modify the manuscript accordingly (see our responses to the general and specific comments). There are however a few points where we share a slightly different point of view, and we believe that there are worth discussing before addressing the comments.

Our main goal with this article is to provide the broader snow community with a simple understanding on why :
- The macroscopic vapor flux in snow cannot be larger than in free air
- Surface kinetics plays a determinant role in the macroscopic vapor flux (and should not be overlooked)
- Previous theoretical and modeling studies erred when they reported a macroscopic vapor flux larger than in free air

Accordingly and  in an effort of simplicity and clarity, we have emphasized physical reasoning and narrowed our focus to the magnitude of the macroscopic vapor flux, rather than using a proper upscaling method and deriving upscaled equations.

On the other hand, we understand from the review that the reviewer considers that the article could benefit from:
- The use of a proper upscaling method (such as the asymptotic-scale expansion or the volume averaging method).
- A restructuring of the article, with a shorter discussion of the hand-to-hand mechanism
- A more more in details study of non-linear kinetics

Our point of view on these points is that:

- We were not able to derive an upscaled equation of the vapor diffusion equation under arbitrary kinetics. From our understanding this is an active topic of research and methods that apply to snow under arbitrary kinetics are not yet readily available. We are currently (with other colleagues) trying to extend the asymptotic-scale expansion method to arbitrary kinetics in snow, but cannot guarantee when and if it will succeed. Therefore, as proposed in our discussion section we see this study as a preliminary work justifying the need for a rigorously upscaled model under arbitrary kinetics.
We will modify Section 3 to better explain that our only goal is to quantify the macroscopic flux, and not to derive upscaled equations.

- We believe that it is necessary to keep a detailed discussion of the hand-to-hand mechanism, as it is a commonly accepted concept in the broader snow community. If the hand-to-hand mechanism were not addressed up front, one could potentially attack the rest of our results by invoking the absence of the hand-to-hand mechanism in our computations.
Similarly a discussion on what is the macroscopic flux and how it relates to microscopic fluxes in the pores is necessary, as it explains most of the discrepancies with previous theoretical studies. We will emphasize that the definition of the macroscopic flux used in this article is physically sound and matches with the definitions used in previously published studies.

- When we introduced the notion of non-linear kinetics, we had in mind to only present a purely illustrative example of the potential impact of non-linear kinetics, and to show that it does not

induce a vapor flux larger than in free air. While we fully agree that an in-depth study of the effects of non-linear kinetics would be more than welcome, we do not think that our article is the right place for it. Indeed, we fear that (i) it will blur our main goals (a clear rebuttal of the hand-to-hand mechanism and of the notion of macroscopically enhanced diffusion in snow) and (ii) at this point we could only offer a superficial look at the matter. A dedicated study, starting on simple structures to decipher the patterns of surface vapor saturation and extending to more complex snow microstructures could be done in the future. One would also need a more detailed treatment of the ice crystals' surface physics (presence of vicinal surfaces, potential non-symmetry of the evaporation and deposition process) to be able to offer quantitative conclusions.

We have copied the comments of the reviewer below in light blue, and provided our answers in black below them. Modifications to the text of the manuscript are proposed in green.

Finally, note that we have discovered two numerical errors in the previous version of the manuscript that we intend to correct in the new version:
- **L343**: The average gradient in the air is 79.00 K/m (and not 77.57 K/m as previously stated).
- **Table 1** and **Fig 7**: The effective diffusion coefficients for the Melt Forms sampled have been underestimated by 33%.

**GENERAL COMMENTS :**

1 – The chosen upscaling method of 'volumetric averaging' over 'cross-section averaging' (l.120 -l.132) is based on the argument that microscopic scale variations are not accessible by area averaging. To my understanding this is an issue related to the Representative Element Volume (REV). Snow microstructures are measured with µCT, large enough such that the volume is representative and homogeneous in a volumetric manner. If cross-sections are used this might not be satisfied anymore as rightly addressed by Pinzer et al. [2012] and volumetric averaging can be chosen. It is therefore not the intrinsically preferred method, but one that is dictated by the specific microstructure.

We agree that computing the macroscopic flux by volume averaging instead of cross-section averaging is related to the specificity of the microstructure. As snow is a random porous medium we believe that cross-section averaging should yield adequate results as long as one select a "large enough" surface to be somehow representative of the overall microstructure. In the case of idealized and periodic microstructure however, the existence of such representative surface is not guaranteed. For instance in a disconnected sphere structure, if one selects an infinitely large cross-section between two planes of spheres, this surface would not be representative and would over-estimate the vapor flux. For this we believe that volume averaging in the safest choice, as it works in situations where cross-section averaging might fail.

We will propose to rewrite the paragraph to indicate that while surface averaging might sometimes be appropriate, volume averaging is a safer choice in general. We will add **L124**:
"Yet, this method of computing the macroscopic vapor flux can be problematic. Indeed, as pointed out by Pinzer et al. (2012) the water vapor fluxes through different horizontal planes of a microstructure are not necessarily all equal. Thus, depending on the chosen plane, the same snow sample could be assigned different macroscopic fluxes, contrary to the notion that the snow sample is homogeneous from the macroscopic point of view. To avoid this issue, the macroscopic flux should therefore be computed as the volume average microscopic vapor flux over the entire representative volume of the microstructure (Shertzer and Adams, 2018), which is equivalent to averaging the fluxes through various horizontal planes (Pinzer et al., 2012)."

2 – The chosen upscaling method is important especially if we couple the effective diffusion to the macroscopic mass and heat transport Calonne et al. [2014]. This study should explicitly relate its results to this study, and how these equations should be adapted.

We do not use a proper upscaling method per se. Contrary to Calonne et al. (2014), we therefore do not attempt to derive the equation governing water vapor at the macroscopic scale.

We will put clearly in the text that we do not upscale to a macroscopic equation and that our goal is only to quantify the macroscopic vapor flux and its dependence on surface kinetics. Therefore, contrary to Calonne et al. (2014) we do not provide a macroscopic vapor equation. We will add a paragraph **L144**:
"Note that the goal of this work is only to quantify the macroscopic water vapor flux in snow and its associated effective diffusion coefficient. Contrary to Calonne et al. (2014) we do not attempt to derive the macroscopic equations governing water vapor at the layer scale."

We also propose to rename Section 4 of the article from "**Quantifying the macroscopic vapor flux in snow**" to "**Bounding the effective diffusion coefficient of water vapor in snow**"

The asymptotic scale derivation of Calonne et al (2014) assumes a small influence of vapor sinks and sources at the ice/pore interface. There is no guarantee that their upscaled equation applies for large α, when sublimation/deposition effect become significant. As a matter of fact, the macroscopic vapor flux that we derive for large α does not correspond to the upscaled model of Calonne et al (2014), as they predict that the vapor flux should be the same as in an inert medium.

3 – The accommodation coefficient, including its name, should be introduced in the introduction including experimental observations such as Libbrecht [2005], Harrington et al. [2019] and possibly other studies. The choice of values for the simulations should be linked and/or motivated by deficiencies of these studies.

We will rename α the sticking coefficient for the entire article.

We will discuss the potential influence of kinetics in the introduction. However, we do not think the sticking coefficient should be explicitly discussed in the introduction. Indeed, experimental measurement of α are mainly limited to the problem of deposition of facets, and it is not a given that such measurements apply for the entirety of ice surfaces in snow, which includes sublimation and non-faceted surfaces. Moreover, this is in line with our methodology to cover a broad range of sticking coefficient values.

We will rewrite the paragraph starting **L18**:
"The physics at play in the pores is generally agreed upon, even though questions about the precise kinetics of the sublimation and deposition of water molecules onto ice surfaces in snow remains open (Legagneux and Domine, 2005, Pinzer et al., 2012, Calonne et al., 2014, Krol and Löwe, 2016). However, even for investigators assuming the same physics at the microscopic scale, the transition from the microscopic to the macroscopic scale remains a point of contention in the snow community (Giddings and Lachapelle 1962, Colbeck 1993, Pinzer et al., 2012, Hansen and Folsien, 2015, Shertzer and Adams 2018)."

We will also rewrite the end of the introduction to better introduce our work on the accommodation coefficient, starting from **L39**:
"The aim of this paper is to clarify the origin of these discrepancies and to quantify the macroscopic vapor flux based on theoretical and numerical modeling. As the kinetics of sublimation and deposition of water molecules on the ice surfaces in snow is not well constrained, we decided to

explore a broad range of possible kinetics in our study. We start by considering in Section 2 whether the hand-to-hand mechanism, as originally proposed by Yosida et al. (1955), can indeed explain the large macroscopic vapor fluxes observed in snow. Then in Section 3, we recall how the macroscopic vapor flux can be obtained from the microscopic vapor flux occurring at the pore scale. In Section 4 we present theoretical work to bound the macroscopic vapor flux in snow, by treating two limiting cases of surface kinetics."

4 – Although symbols in equations are generally well described and it is clear from the context what they mean, it might be helpful to the reader to introduce systematic notation to distinguish between upscaled quantities and local quantities, e.g.

$$F = \frac{1}{V} \int_V f \, dx^3$$

in other words, how are F and C related to their microscopic quantities?

We will add in the text **L132** the equation giving the macroscopic water vapor flux from the microscopic water vapor flux.

"Again, the averaging needs to be performed over the total volume, including the ice space, and the macroscopic vapor flux F is thus given by

$$F = \frac{1}{V} \int_{V_a} f \, dV$$

where V and $V_a$ respectively represent the total volume of the snow sample, and the pore volume."

5 – In case of volume averaging, gradients of microscopic fluxes are influenced by sources and sinks at internal ice-air iterfaces Whitaker [1998], Krol and Löwe [2018], i.e.

$$\langle \nabla f \rangle = \nabla \langle f \rangle + \int f \, d\boldsymbol{n}$$

In case of the idealized spheres the second term vanishes because of symmetry, but for your snow samples it might not be the case, and should be shown, either by estimating the order of magnitude of the gradient of your sources and sinks, or by analysis of the simulations that this term is rightfully neglected. Here it matters how the macroscopic quantities are related to their microscopic counterparts. Note that in your simulations you average over both phases, vapor and ice, but you neglect the sinks and sources. I believe with these microstructures it is probably alright, but it should be estimated/shown that you can do so.

We do not use the spatial averaging theorem (SAT) in our treatment, and directly compute the left-hand side of the equation, integrating the microscopic vapor fluxes over the microstructure. The effects of sinks and sources at the ice/pore interface are taken into account directly at the microscopic scale, through Robin or Dirichlet boundary conditions.

We will however include a new appendix using the SAT to demonstrate that the macroscopic flux is maximal in the infinitely fast kinetics case. This new appendix is attached at the end of this document.

**SPECIFIC COMMENTS:**

l10: Naming of the coefficient α. This coefficient is often related to the phase-change it represents i.e. deposition, sublimation, or sticking parameter.

We will use replace "accommodation coefficient" with "sticking coefficient" throughout the manuscript.

l11: There is no evidence or discussion in the paper that suggests that convection is one of the candidates responsible for the experimentally observed mass deficiency.

We will remove the mention of convection from the abstract, and rewrite the discussion **L437**: "Indeed, the importance of convective mass transport in subarctic snowpacks has notably been pointed out by Trabant and Benson (1972) and Sturm and Johnson (1991), and thus appears as a good candidate to explain the high vapor movement in subarctic snowpacks."

l46: Suggestion to shorten this paragraph and move to the discussion. The notion of hand-to-hand diffusion should be discarded on the fact this is simply no physical transport of water molecules.

We think it is important to keep this full paragraph. Indeed, the notion of water molecules short-cutting the ice space is commonly held in the snow science community, and we think a detailed rebuttal is welcome. As a concrete example, when we initiated this study we thought that the hand to hand mechanism as described by Yosida et al. (1955) was a valid mechanism.

l120-132: Please be very specific about your methods of upscaling. See general comments 1, 2, and 5.

As mentioned in the general comments 2 and 5, we do not use any method of upscaling and simply directly compute the volume average of the microscopic vapor flux. Because of it, we cannot provide an upscaled equation. We instead rely on a phenomenological approach.

We however added precision that in our numerical and theoretical computations we assumed that the macroscopic gradient of water vapor equals the macroscopic gradient of saturated vapor.

For this we added **L103**:
"Let us consider a volume of snow (Figure 2a), subjected to vertical macroscopic temperature and vapor gradients at its boundaries. For this study we consider that the macroscopic water vapor gradient equals the macroscopic gradient of saturated vapor, and is therefore driven by the macroscopic temperature gradient (as in Yosida et al.,1955, Colbeck, 1993, Sokratov and Maneo, 2000, Pinzer et al., 2012)."

and **L141:**
"However, one should keep in mind that the effective diffusion coefficients computed in this work might depend on the applied vapor and thermal gradients, and are therefore not necessarily intrinsic. Moreover the proposed numerical values may also not apply in the case where the macroscopic concentration gradient is decoupled from the macroscopic thermal gradient."

We will also add a discussion on this assumption **L435**:
"Furthermore, we assumed in this study that the macroscopic water vapor gradient is equal to the macroscopic gradient of saturated vapor, driven by the macroscopic thermal gradient. This assumption has been regularly made in the snow science community (Yosida et al.,1955, Colbeck, 1993, Sokratov and Maneo, 2000, Pinzer et al., 2012), and is supported by the idea that the ice in

the snowpack tends to impose water vapor saturation at the macroscopic scale. It however remains possible that the macroscopic water concentration deviates from saturation, notably if the deposition and sublimation kinetics is slow. A rigorous upscaling method yielding the equations governing macroscopic water concentration would therefore also help quantifying if such a situation of non-saturation at the macroscopic scale is likely to occurs in real snowpacks, and indicate how the macroscopic vapor flux should be computed in such a case."

L133: Here I would expect a mathematical definition, including upscaling methods, see comment 5.

As explained above, we directly compute the macroscopic vapor flux as the volume average of the microscopic vapor flux (and justify this choice with physical consideration). We will add reference to previously published study of Schertzer and Adams (2018) that uses the same formula for the computation for the macroscopic flux. We will write the equation relating the macroscopic water vapor to the microscopic one (see our proposed text modification in the response to the general comment 4).

L137: Semantic comment: What does 'ideally' mean in this context? Maybe include that intrinsic, in this context, means that $D_{eff}$ is independent of the external temperature gradient. When $D_{eff}$ is dependent on the external gradient, one could say that the response of the material is non-linear. Does this break the definition of the effective diffusion coefficient, meaning the coefficient that quantifies the vapor flux as a linear response to an applied concentration gradient?

In general, in the snow community, when one uses an effective diffusion coefficient one expects it to be independent of the applied concentration gradient and thus to be a true proportionality constant between flux and gradient. While standard upscaling techniques such as the asymptotic-scale expansion or the volume averaging method prove that the obtained diffusion coefficients are independent of the applied gradient (under certain conditions of course), our phenomenological approach does not prove it. We therefore want to stress that what we defined as the effective diffusion coefficient might not be intrinsic (even-though it appears to be the case with linear kinetics based on our simulations).

In our article, we call the ratio of the flux over the gradient the effective diffusion coefficient, even in the case where it is not constant with respect to the gradient. While we understand that some might prefer to use a different term in this case, we choose this approach as it allow us to easily compare it to the diffusion coefficient in air and is easily understood by the broader snow science community. We will rephrase **L137** with:
"In the snow science community the effective diffusion coefficient $D_{eff}$ is usually expected to be independent of the applied thermal and vapor gradients (e.g. Yosida et al., 1955, Colbeck, 1993). In this case, it is possible to treat the problem of macroscopic vapor transport in snow with a generalized Fick's law, where $D_{eff}$ is independent of the applied boundary conditions and only depends on the snow microstructure. Such an effective diffusion coefficient does not depend on the external conditions, and is then said to be intrinsic (Auriault et al., 2010). However, one should keep in mind that the effective diffusion coefficients computed in this work might depend on the applied vapor and thermal gradients, and are therefore not necessarily intrinsic."

L150: In this paragraph I suspect at least an expression for the the macroscopic vapor flux as suggested by the title.

We will explain in the text that solving the microscopic equations yield the microscopic water vapor fluxes and that volume averaging the microscopic fluxes yields the macroscopic water vapor flux:
"Solving Equation 2 we obtain the microscopic vapor fluxes inside the whole microstructure. Using Equation 1 then yields the water vapor flux at the macroscopic scale F."

L165 and 308: How infinite can α $v_{kin}$ be? $v_{kin}$ is finite $\sim 10^2$ , and $0 < α < 1$. In principle it should be compared to the actual interface velocity v n in the Robin b.c. as stated in Kaempfer and Plapp [2009]. A discussion on α and its values would be appreciated Libbrecht [2005], Saito [1996], Legagneux and Dominé [2005].

The notion of infinitely fast kinetics is purely theoretical, and should be seen as a limiting case. We use this notion to treat the case where the kinetics is fast enough to impose vapor saturation at the interface. In practice, α $v_{kin}$ cannot indeed be greater than vkin. We will clarify in the text, that treating α $v_{kin}$ as infinite is a simplifying assumption. The validity of this assumption to model macroscopic vapor diffusion in snow is treated in the discussion, and remains an open question. We will add **L165**:

"While the infinitely fast kinetics case is strictly theoretical, as α $v_{kin}$ is less than or equal to $v_{kin}$, it helps apprehending the macroscopic vapor flux when surfaces kinetics processes are much faster than diffusion in the air space"

To know if the infinitely fast kinetics is appropriate for macroscopic flux modeling, we believe that one should compute the value of the second Damkohler number (defined as Da = α * $v_{kin}$ * l / $D_0$), where l is a length characterizing the microscopic scale. Such an approach is notably used in the diffusion-reaction community (e.g. Munnichi and Icardi, 2020 and Bourbatache et al., 2020). This number essentially compares the characteristic times of the surface processes and of diffusion in the air space, and characterize how the oversaturation at the ice/pore interface compares to the concentration gradient in the pores.

We do not believe that the interface velocity in itself is a good quantify to differentiate the fast and slow kinetics regime of the macroscopic vapor flux. Indeed, increasing the thermal gradient at constant α increases the interface velocity, but do not necessarily modify the effective diffusion coefficient, as seen in our Figures 4 and 6 for example.

L263: This paragraph includes an important realization, how does it relate to the expression for the macroscopic heat transport provided by Calonne et al. [2014]. This could be treated in the discussion.

The study of Calonne et al. (2014) is based on the assumption of slow kinetics. On the other hand, Hansen and Folsien (2015) assume that water vapor is constantly saturated at the microscopic scale, which corresponds to infinitely fast surface kinetics. Therefore, while Calonne et al. (2014) find that heat conduction in snow occurs similarly as in an inert medium, Hansen and Folsien (2015) find that water vapor becomes an integral part of heat transport with fast kinetics.

The degree of coupling between heat transfer depends on the kinetics of sublimation and deposition. We are currently working on a manuscript investigating the different behavior between very slow and very fast kinetics (thus not investigating the intermediate cases and the transition between slow to fast kinetics). Our preliminary results are in line with the idea that with very fast kinetics, vapor transport becomes an integral part of heat transfer (as proposed by Yosida et al., 1955 or Moyne et al., 1988) and appears in the effective thermal conductivity of snow.

We will add in the text **L264** that:
"In their model, water vapor in at constant saturation in the pores (thus corresponding to the case of infinitely fast kinetics), and acts an integral part of heat transfer by transporting latent heat between sublimation and deposition surfaces (as notably proposed by Yosida et al., 1955)"

L282: Some more details on the technicalities of the simulation should be provided, are T and c computed simultaneously? or is c computed given T ? How is it parallelized, and how long does it take? What are the meshing requirements, how many points etc.

We will extend the paragraph **L330** to add more details on how the simulations are performed. "The heat and diffusion equations are solved using the finite element method with the open-source software ElmerFEM (Malinen and Raback, 2013). We use the readily available ElmerFEM modules dedicated to the heat and diffusion equations, which are solved with iterative methods. We first solve the steady-state heat equation in order to obtain the temperature field in the entire microstructure. The steady-state vapor diffusion equation is then solved using the saturation concentration at the ice/pore interface resulting from the previously computed temperature field. In the case of simulations performed on measured snow microstructures, the tetrahedral meshes have been derived from Xray computed microtomography images using the CGAL meshing library. The meshes have been refined to capture the ice/pore interface, and contains between 18 and 50 millions elements, depending on the snow sample. Moreover, in the case of snow samples the meshes have been partitioned into 20 sub-meshes and the computation are performed using the parallel computing abilities of ElmerFEM. Under such conditions, a simulation typically takes a bit less than an hour to run. Finally, the outputs of the simulations are processed using the ParaView software to compute the volume averages."

L367: For the non-linear kinetics results it might be useful to state the surface averaged simulated α and its variance.

We computed the surface averaged α as well as the associated standard deviation, for the DF snow sample. We find that the average value increases with temperature gradient, from 0.00145 at 5 K/m to 0.00204 at 200 K/m. Note that these value do not align with the constant α yielding equivalent macroscopic vapor fluxes.
Moreover, the variance remains high in all cases with a value around 0.029. Expressed relatively to the average value, the relative standard deviation drops from 20 at 5 K/m  to 14 at 200 K/m. This high variability is consistent with observations that values of α cover almost the entire 0 to 1 interval within the microstructure.

[Figure]

We have included below two Figures showing α and the relative standard deviation as a function of thermal gradient.

[Figure]

Our understanding is that these results indicate that in the case of non-linear kinetics, the macroscopic flux is driven by very localized effects, not easily accessible through average quantities. Clearly this is an interesting topic of research, but as explained in the introduction of our response we believe that a dedicated and more thorough study would be more appropriate to discuss these points in details. Moreover, as we are not confident in the validity of the non-linear law chosen in this article, we should be careful not to over interpret these preliminary results.

L367: How sensitive is your result to the value $\sigma_0$ ? Since it might differ for different crystallographic surfaces.

We did not performed simulations with varying $\sigma_0$ parameters. Our understanding is that increasing $\sigma_0$ should shift the green curves of Figure 4 and 6 towards the right, as greater saturations, and thus greater macroscopic concentration gradients are needed to have high $\alpha$.

However, as we are not sure that the used formulation of $\alpha$ as a function of saturation applies for the entirety of ice surfaces in snow, our results on non-linear kinetics should only be viewed as quantitative at this point.

Fig 4: I suggest to split this plot into two figures. One for linear simulations $D_{eff}^{norm}$ vs $\alpha$ and the other for non-linear dynamics $D_{eff}^{norm}$ vs $\nabla T$ including colorbar for surface averaged $\alpha$. This suggestion is given to observe the type of transition between purely tortuous diffusion and phase transition enhanced diffusion. The data on the non-linear dynamics seems to rapidly depart from the tortuous diffusion case: is there a reason for this? We would expect also here a smoother transition between the two limiting cases, such as in Fig.6. The results for small temperature gradients puzzle me. A discussion on the results in this regime might be helpful.

We do not think our results for non-linear kinetics should be further interpreted in this paper. As explained before, we are not confident that the chosen law applies for the entirety of the snow sample.

We do not know why there is a faster transition from the tortuous diffusion to the phase transition enhanced diffusion for the idealized microstructure than for the measured microstructure. To answer this question a potential method would be to compare the results obtained on various simple

idealized microstructure, to decipher the influence of porosity and tortuosity, and quantify where the zones of deposition and sublimation appear in the microstructure.

L402: Moreover? Is there a reason not to compute the non-linear cases? In my opinion it is interesting and worth it to quantify the different non-linear responses of the 6 different snow types.

As we are not sure that the chosen formulation for non-linear kinetics applies to snow, our goal is only to provide an illustration of the effect of non-linear kinetics. We illustrate that it does not produce greater than in air vapor fluxes, but cannot produce more quantitative results. While it is certainly an interesting topic, it is out of our scope for now and we are not able to reach robust conclusions concerning the effects of non-linear kinetics.  We will add to the text **L355**:
"[…] and might not properly apply for the entirety of ice surfaces in snowpacks. Indeed, this law has been derived using deposition measurement, and might not apply for sublimation surfaces (Beckmann and Lacmann, 1982). Moreover, we cannot rule out the presence of vicinal surfaces in the snowpack, where the proposed law does not apply (Legagneux and Domine, 2005). Therefore, the point of using such a law is to qualitatively study the potential impact of a dependence of α to the local vapor saturation, rather than to produce quantitative results."

We will also explain why we chose not to perform non-linear kinetics simulations on all our sample set:
"We also did not compute $D_{eff}^{norm}$ with non-linear surface kinetics (i.e. when α is not constant), as we are not confident in the validity of the chosen non-linear law for snow modeling."

Fig 7 :  The Figure and Table have approximately the same information. Consider plotting again $D_{eff}$ vs α and colorbar on density. Alternative, plot $D_{eff}/\varphi$ and discuss the remaining influence of SSA. If SSA is presented in either a table or a plot, then a note in the discussion on its influence is desirable.

Our reasoning behind putting both the Figure and the Table is that the Figure helps apprehending the influence of density and α, while the Table provides the exact data point.

We have redone the Figure to plot $D_{eff}$ as a function of α (inserted below in our response). This new figure clearly highlights that there are two regimes for the effective diffusion coefficient, with a transition for constant α coefficients around 1e-3.

We will rewrite the sentence **L404** to:

[Figure]

"Figure 7 highlights that the normalized effective diffusion coefficient exhibits two different regimes depending on the value of α. The transition between the fast and slow surface kinetics regimes occurs for values of α around $10^{-3}$."

It is hard to decipher the potential influence of SSA, as in our sample set SSA is well correlated with density. We will emphasis this point in the manuscript **L405**:
"We observe that the effective diffusion coefficient is well correlated with density, and show an almost systematic decrease of $D_{eff}^{norm}$ with increasing density, for all values of α. The correlation between $D_{eff}^{norm}$ and the specific surface area is not so well marked, notably for the RG sample that shows a large value of specific surface area without any clear impact on $D_{eff}^{norm}$. That being said, our sample set is only composed of six samples and for which density and specific surface area are correlated. A detailed study of the influence of microstructural parameters on the effective diffusion coefficient would require a larger sample set, notably to be able to decipher the independent influence of specific surface area and density."

L440: A list of the general causes to why vapor flux was considered to be enhanced in the past is expected in the discussion.

We will add a new sentence at the end of the first paragraph of the discussion **L411**:
"We have shown that most of the previous theoretical studies reporting macroscopic vapor flux enhanced compared to free air used faulty computations of the macroscopic vapor flux, which resulted in systematic overestimation."

L400: A reasonable explanation for why convection could be the cause of the experimentally observed mass deficit could go here.

We will reformulate the discussion on convection to **L437**:
"Indeed, the importance of convective mass transport in subarctic snowpacks has notably been pointed out by Trabant and Benson (1972) and Sturm and Johnson (1991), and thus appears as a good candidate to explain the high vapor movement in subarctic snowpacks."

L448: 'Disagree', is an understatement. You show with numerical simulation that this concept is ill-defined. Suggestion: We show with numerical simulations that increased vapor flux by the hand-to-hand mechanism is not present.

We do not think our simulations can be used to show that the concept of hand-to-hand diffusion is ill-defined, as they by default do not include the hand-to-hand mechanism. As stated in the introduction to the review we believe that Section 2 and specific physical reasoning is necessary to refute the hand-to-hand mechanism.

L458: Avoid 'intuitive'. Suggestion: consistent with actual water vapor transport.

We will rephrase to "We argue that the method used in this article, i.e. volume averaging over an entire microstructure including the ice, is the only one consistent what is the actual nature of the macroscopic water vapor flux."

L492: incorrect use of 'inferior', use 'less than'.

We will systematically replace "superior" by "greater than" and "inferior" by "less than".

**TECHNICAL CORRECTIONS:**

L59: The use of pore phase, throughout the manuscript is incorrect. Please use pore space, or vapor/gas phase. Also air phase is not commonly used.

We will rewrite the manuscript using the terms "pore space" and "gas phase".

Overall The use of colons is not consistent, e.g. before equations introduced by 'given by' it is not very common to use them. Use of colons is generally restricted to lists or 'may' be used between independent clauses when the second sentence explains, illustrates, paraphrases, or expands on the first sentence. Equations are part of sentences and therefore colons should not appear more often before an equation than in other parts of your text.

We will remove the colon before the introduction of the equations

l.296 Outer brackets in the exponent should be larger, (use \left( and \right) commands).

We will rewrite the equation with larger brackets.

l.304 Condensation is reserved for the gas-liquid phase-transition, use deposition (or desublimation) also at other places throughout the manuscript.

We will replace condensation with deposition throughout the manuscript.

l.336 Goes → go.

We will correct the typo.

l.437 ? citation missing.

Yes a citation referring to to Sturm and Benson 1997 was missing. We will fix it in the new manuscript.

l.454 Similar → Equivalent.

We will rephrase as proposed.

**Demonstration that the macroscopic vapor flux is maximal under infinitely fast kinetics**

The aim of this appendix is to demonstrate that the macroscopic vapor flux is maximal in the case of infinitely fast kinetics. For this we start by applying the spatial averaging theorem whitaker1999method to the vapor concentration in the pores $c$

$$< \nabla c >= \nabla < c > + \frac{1}{V} \int_\Gamma c \, \mathbf{n} \, \mathrm{d}S \qquad (1)$$

where $< \bullet >$ is an operator defined as $< \bullet >= \frac{1}{V} \int_{V_\mathrm{a}} \bullet \, \mathrm{d}V$, and the concentration $c$ in the surface integral is the vapor concentration at the ice/pore interface. Multiplying by $D_0$, and using the notation introduced in this article for the macroscopic vapor flux $\mathbf{F}$, we have

$$\mathbf{F} = -D_0 \nabla < c > - \frac{D_0}{V} \int_\Gamma c \, \mathbf{n} \, \mathrm{d}S \qquad (2)$$

Moreover, using the Hertz-Knudsen equation we have that the concentration at the interface is

$$c = c_\mathrm{sat} - \frac{D_0}{\alpha v_\mathrm{kin}} \nabla c \cdot \mathbf{n} \qquad (3)$$

Equation 2 can thus be written as

$$\mathbf{F} = -D_0 \nabla < c > - \frac{D_0}{V} \int_\Gamma c_\mathrm{sat} \, \mathbf{n} \, \mathrm{d}S + \frac{D_0^2}{V \alpha v_\mathrm{kin}} \int_\Gamma (\nabla c \cdot \mathbf{n}) \, \mathbf{n} \, \mathrm{d}S \qquad (4)$$

Applying the same spatial averaging theorem to the saturation concentration $c_\mathrm{sat}$, we have

$$\frac{1}{V} \int_\Gamma c_\mathrm{sat} \, \mathbf{n} \, \mathrm{d}S =< \nabla c_\mathrm{sat} > - \nabla < c_\mathrm{sat} > \qquad (5)$$

Injecting Equation 5 in Equation 4 thus yields

$$\mathbf{F} = -D_0 \nabla < c > - D_0 < \nabla c_\mathrm{sat} > + D_0 \nabla < c_\mathrm{sat} > + \frac{D_0^2}{V \alpha v_\mathrm{kin}} \int_\Gamma (\nabla c \cdot \mathbf{n}) \, \mathbf{n} \, \mathrm{d}S \qquad (6)$$

As we assume that the macroscopic vapor concentration equals the macroscopic saturation concentration gradient (as in [5, 1, 3, 2]), we have that $\nabla < c >= \nabla < c_\mathrm{sat} >$. Thus

$$\mathbf{F} = -D_0 < \nabla c_\mathrm{sat} > + \frac{D_0^2}{V \alpha v_\mathrm{kin}} \int_\Gamma (\nabla c \cdot \mathbf{n}) \, \mathbf{n} \, \mathrm{d}S \qquad (7)$$

Let us now assume, without loss of generality, that the macroscopic vapor and thermal gradients are orientated downward. As seen in Figure 1, surfaces that are characterized by a normal vector pointing upward are deposition surfaces. The product $\nabla c \cdot \mathbf{n}$ is therefore negative, and $(\nabla c \cdot \mathbf{n}) \, \mathbf{n}$ is a vector pointing downward. Similarly, surfaces that are characterized by a normal vector pointing downward are sublimation surfaces. The product $\nabla c \cdot \mathbf{n}$ is thus positive, and

[Figure]

Figure 1: Schematic showing the normal vector **n** of deposition and sublimation surfaces. Ice crystals are represented in blue, and the thermal and vapor gradients are assumed to point downward.

the vector $(\nabla c \cdot \mathbf{n})\,\mathbf{n}$ is pointing downward. Therefore, for both type of surfaces $(\nabla c \cdot \mathbf{n})\mathbf{n}$ is pointing downward. The surface integral term in Equation 7 thus acts in opposition of $-<D_0 \nabla c_{\text{sat}}>$, and tends to reduce the macroscopic vapor flux. We thus have the inequality

$$|\mathbf{F}| \leq |<D_0 \nabla c_{\text{sat}}>|  \tag{8}$$

We will now show that this upper bound is reached in the infinitely fast kinetics case. Indeed, under the infinitely fast kinetics hypothesis the product $\alpha v_{\text{kin}}$ can be treated as going to infinity. At the same time, the surface integral of Equation 7 remains bounded, as the concentration gradient in the vicinity of the interface does not diverge. The surface integral thus vanishes, and the norm of the vapor flux is given by

$$|\mathbf{F}| = |<D_0 \nabla c_{\text{sat}}>|  \tag{9}$$

that is to say that the upper bound of the macroscopic vapor flux is reached under infinitely fast kinetics. Moreover, note that we re-derived that in the infinitely fast kinetics case, the macroscopic vapor flux is given by the spatial average of the saturation vapor concentration in the pore space.

**References**

[1] S. C. COLBECK, *The vapor diffusion coefficient for snow*, Water Resour. Res., 29 (1993), pp. 109–115.

[2] B. R. PINZER, M. SCHNEEBELI, AND T. U. KAEMPFER, *Vapor flux and recrystallization during dry snow metamorphism under a steady temperature gradient as observed by time-lapse micro-tomography*, Cryosphere, 6 (2012), pp. 1141–1155.

[3] S. A. SOKRATOV AND N. MAENO, *Effective water vapor diffusion coefficient of snow under a temperature gradient*, Water Resour. Res., 36 (2000), pp. 1269–1276.

[4] S. Whitaker, *The method of volume averaging*, vol. 13, Springer Netherlands, 1999.

[5] Z. Yosida, H. Oura, D. Kuroiwa, T. Huzioka, k. Kojima, S.-I. Aoki, and S. Kinosita, *Physical studies on deposited snow. I. Thermal Properties.*, Contributions from the Institute of Low Temperature Science, 7 (1955), pp. 19–74.

---

## Author Response (AR1)

**TC-2020-183**

**RESPONSE TO THE EDITOR**

Dear Editor,

Thanks for your reactivity on the editing process. A revised version of our manuscript has been uploaded on The Cryosphere's portal.

We also attached at the end of this document a version of the manuscript, with the differences between the new previous versions highlighted. Removed text is shown as red strikethrough and added text in underlined blue. Please note that some of the added or removed citations are not properly handled in the difference document, and therefore sometimes do not fit entirely on the page (for instance L21 and L23 below). These citations are however properly displayed on the revised version of the manuscript, without the highlighted differences.

Best Regards,
Kévin Fourteau, on behalf of all co-authors

[revised manuscript text omitted]

---

## Referee Report (RR1)

**Review 2: Macroscopic water vapor diffusion is not enhanced in snow**

Dear Authors,

The revised version of this paper has significantly improved over the first version and I thank the authors for incorporating the necessary improvements suggested by Dr. Kevin Hammonds and me. The authors have chosen to focus on arguing and showing that the phenomenological vapor transport is not enhanced. In view of this goal I think the authors have done an excellent job, especially on the discussion of their work in relation to existing literature.

A few suggestions:

- Since it is a controversial topic it is important to be as clear as possible about how this work relates to previous works, including the TCD paper of Andrew Hanson Hansen [2019]. This paper has been part of the scientific attempts to explain vapor transport in snow. I would suggest to include a small discussion on this paper in the introduction or in the discussion.

- Consider also discussing **?**. They use higher values of $D_{\text{eff}}$, how would their results change? Do they have to redo their simulations?

- l.85, The question related to the Yosida experiment might be the following: Is Weighing the different compartments not a flawed way of measuring the total vapor flux? For this you have to know the exact ice matrix. Because vapor and advective ice mass are connected, the advection of the ice matrix in the opposite direction should be subtracted. That the advective ice is important is also given by the alternative prediction of the vapor flux in **?**, where they equate the advective ice to the opposite vapor flux and get a an estimate that is close to the flux which was found based on the FE-based saturated vapor concentration flux.

- l.86, The word 'tempted' is suggestive, consider: If you adopt the hand-to-hand mechanism such as Hansen [2019] and . . . . Be specific rather then suggestive.

- l.104. suggestion: 'and in particular if they are, on average, larger than . . . '

- Consider including appendix A inside your main text. First it proves that $D_{\text{eff}}$ is maximal under infinitely fast kinetics, and is in fact equal to completely saturated condition (In the robin boundary condition such as used in Kaempfer and Plapp [2009] this limit is well defined). Then you proof that it is smaller than equation 8.

- The parameter $\phi$ is usually assigned to the ice volume fraction, it might therefore be helpful if you state that when you define it, for example: 'not to be confused with ', and restate that in your appendix when you re-derive the equations of Hansen and Foslien [2015].

- Appendix C: I did not have time to check all the volume fraction terms in comparison with Hansen and Foslien [2015], sorry for that. Maybe double check that there are no mistakes there. One question here: why is the tabular flux weighted by the ice phase and the laminar one by the vapor? Also it might be interesting to check what the origin is of this $\phi$ difference term, is it only the hand-to-hand mechanism? or does it come from different definitions of averaging?

- The brackets $<,>$ look funny, rather use $\langle c \rangle$. (\left > and \left >).

- word usage: preponderant? Use simpler words whenever possible. Consider deleting this sentence, since this is discussed in your paper, but is not a result of your paper.

- The general word 'indeed' sounds very odd in a few places and is sometimes confusing, and rarely used in the beginning of a sentence. Please check the meaning of these sentences. For example l.494, you can avoid using Indeed, by: "Both Trabant and Benson (1972) and Sturm and Johnson (1991) already pointed out the importance of..."

- l.7 – l.508 condensation is not replaced by deposition as you suggested you would do.

- last sentence of the conclusion: This is way to strong... you can't know that this is the only way, there might be other approaches that you haven't explored, please don't overstate.

In closing I would like to congratulate the authors on a very interesting paper, with kind regards,

Quirine Krol

**References**

A. Hansen. Revisiting the vapor diffusion coefficient in dry snow. *The Cryosphere Discussions*, pages 1–27, July 2019. ISSN 1994-0416. doi: 10.5194/tc-2019-143.

A. C. Hansen and W. E. Foslien. A macroscale mixture theory analysis of deposition and sublimation rates during heat and mass transfer in dry snow. *The Cryosphere*, 9 (5):1857–1878, Sept. 2015. ISSN 1994-0424. doi: 10.5194/tc-9-1857-2015.

T. U. Kaempfer and M. Plapp. Phase-field modeling of dry snow metamorphism. *Phys. Rev. E*, 79(3, Part 1), 2009. ISSN 1539-3755. doi: 10.1103/PhysRevE.79.031502.

---

## Author Response (AR2)

**TC-2020-183: Response to the second review of Quirine Krol**

We thank Quirine Krol for her review of the revised version of our article. A point by point response to her remarks is provided below. Her remarks are reported in blue, with our corresponding response in standard black just below. When we modified the manuscript, the newly proposed version is written in *black italics with yellow highlighting*, with the the line of the modification given in **bold black**. The lines indicated correspond to the track-change manuscript, attached at the end of the document.

The most important revision of the manuscript concerns the Appendix C about the Hansen and Folsien (2015) model. We now provide more details on our reasoning and computations. The final conclusion is the exact same, but readers can now better understand our point of view.

All modifications made to the manuscript are highlighted in a track-change version attached at the end of this document.

Best Regards,
Kévin Fourteau, on behalf of all co-authors.

Since it is a controversial topic it is important to be as clear as possible about how this work relates to previous works, including the TCD paper of Andrew Hanson Hansen [2019]. This paper has been part of the scientific attempts to explain vapor transport in snow. I would suggest to include a small discussion on this paper in the introduction or in the discussion.

From our understanding, the Hansen (2019) TCD paper supports the notion of enhanced diffusion with two main arguments:
- The hand-to-hand mechanism as proposed by Yodisa et al. (1955) and discussed in Section 2 of our article.
- The analytical model first presented in Hansen and Folsien (2015) and discussed in Section 4.3 and Appendix C of our article

We thus added a citation of the Hansen (2019) article in the introduction of the article, among the articles involved in the controversy about the diffusion of water vapor in snow **L23**:
*"However, even for investigators assuming the same physics at the microscopic scale, the transition from the microscopic to the macroscopic scale remains a point of contention in the snow community (Giddings and Lachappelle, 1962, Colbeck, 1993, Pinzer et al., 2012, Hansen and Folsien ,2015, Shertzer and Adals, 2018, Hansen, 2019)."*

We also explicitly cite it as an article supporting the hand-to-hand mechanism in Section 2 of the article **L87**:
*"If one adopts the hand-to-hand mechanism, such as Hansen (2019) for instance, the idea of water vapor shortcutting the ice may appear supported by the indistinguishability of water molecules."*

We added that Hansen (2019) proposes to use the model presented in Hansen and Folsien (2015), when discussing this model **L292**:
*"Finally, Hansen and Folsien (2015) and Hansen (2019) proposed an analytical expression for the effective thermal conductivity of snow, taking into account the latent heat associated with the transport of water vapor."*

Finally, by reading carefully Hansen (2019) paper, we realized that some of our sentences describing the state of the art were somewhat similar to his text. We therefore slightly changed our wording to avoid any ambiguities. These modifications occur

**L28:** *"The seminal study of Yosida et al. (1955) set out to measure in the laboratory [...]"*

**L276:** *"Christon et al. (1994) performed finite element microscale simulations of vapor diffusion in snow under a thermal gradient, using an idealized microstructure."*

Consider also discussing Jafari et al. (2020). They use higher values of D eff , how would their results change? Do they have to redo their simulations?

It is true that Jafari et al. (2020) use an effective diffusion coefficient larger than that of the air, while our article demonstrates it is not possible. We thus think that the Jafari et al. (2020) simulations should use a reduced diffusion coefficient. Reducing the diffusion coefficient of water vapor would probably decrease the simulated mass loss of the basal layers, but only a new set of simulations could definitely answer this point.

We also want to point out that Hansen and Folsien (2015) and our article assume that the vapor is saturated at the macroscopic scale. However, Jafari et al. (2020) report significant degree of under and oversaturation in snow layers. While the potential presence of non-saturated layers in a snowpack is an interesting result, this questions the validity of using diffusion coefficients specifically deduced under vapor saturation.

Moreover, the numerical values provided in the Jafari et al. (2020) article indicate that the source term in their mass conservation equation (Equation 2 in the paper) is computed for a sticking coefficient alpha close to $10^{-7}$. This corresponds to slow kinetics, and we thus think that in this case the article should use the model of Calonne et al. (2014), which applies for such a small sticking coefficient.

We think that to move forward, modeling approaches such as the one proposed by Jafari et al. (2020) would greatly benefit from the rigorous upscaling of mass and heat equations for arbitrary surface kinetics, in order to extend the Calonne et al. (2014) to fast kinetics. It would provide non-ambiguous formulations on:
(i) - How to compute the vapor mass flux in the case of undersaturated/oversaturated snow, and to know if it is simply proportional to the vapor concentration gradient, or if there is an additional flux term driven by the temperature gradient (equivalent to a macroscopic Soret effect).
(ii) – How to derive consistent values of the source and vapor flux terms, that should be both taken with the same sticking coefficient.

A detailed discussion of the Jafari et al. (2020) article, properly addressing the points raised above, would go beyond the scope of our article, that solely discusses the macroscopic vapor fluxes. We therefore do not discuss the Jafari et al. (2020) paper in details, but now explicitly cite it as an example of article using larger than in free air diffusion coefficient **L503**:
*"Currently, detailed snow physics models do not include the mechanism of convective mass transport (Lehning et al., 2002, Vionnet et al., 2012) and assume all mass transport to result from diffusion, sometimes using a diffusion coefficient larger than that in free air (e.g. Jafari et al., 2020)."*

l.85, The question related to the Yosida experiment might be the following: Is Weighing the different compartments not a flawed way of measuring the total vapor flux? For this you have to know the exact ice matrix. Because vapor and advective ice mass are connected, the advection of the ice matrix in the opposite direction should be subtracted. That the advective ice is important is also given by the alternative prediction of the vapor flux in ?, where they equate the advective ice to

the opposite vapor flux and get a an estimate that is close to the flux which was found based on the FE-based saturated vapor concentration flux.

We think that the original idea of Yosida et al. (1955) on how to measure the vapor flux is adequate in principle, as it sets to measure what the vapor flux physically is: the transfer of mass from one control volume to another. We think that the experiment of Yosida et al. (1955) fails because the presence of the meshes perturbs the local microstructure and create gaps between the cans. Such a gap concentrates the thermal gradient without any ice blocking, which means that the vapor flux at the cans' interfaces is much larger than the vapor flux in the undisturbed snow (this argument was notably advanced by Giddings and Lachappelle, 1962).
The experiment of Sokratov and Maeno (2000) is revealing on this point, as they did not use meshes to compartment the snow samples during the diffusion process. This removes the potential presence of gaps, creating large vapor fluxes between the control volumes, and their results indicate vapor fluxes from which a diffusion coefficient lower than that of the air is deduced.

While the apparent advection of the ice structure and the vapor flux are clearly related, we do not think that their magnitude are equal in general, and that it is necessary or sufficient to know one to deduce the other.

First, we do not think it is necessary to know the (apparent) advection of the ice matrix to experimentally measure the vapor fluxes, because the motion of the ice/pore interface corresponds to water molecules changing phases but not being macroscopically transported (the ice appearing below a crystal is composed of molecules already present in the vicinity of the interface and thus do not account for any mass flux). To put it differently, the motion of the ice structures does not correspond to a motion of mass, and can be ignored for macroscopic mass transfer.

We also think that knowing the motion of the ice structure is not sufficient to deduce the vapor flux, without a further assumption on the kinetics and diffusion path of water molecules. To compute the vapor flux from the interface velocity, Pinzer et al. (2012) assumed that all the vapor flux crossing a given horizon deposits on the closest ice crystals above. The underlying assumption is reasonable in the case of fast kinetics, where water molecules tend to jump from one ice crystals to another, without any bypass. However, for slower kinetics vapor bypassing ice crystals becomes common, which breaks the assumption and the methodology of Pinzer et al. (2012). For instance, in the case of a very slow kinetics the motion of the interface is close to zero, but one cannot deduces that the macroscopic vapor flux is also vanishing.

Given these consideration, we felt it was not useful to make any addition to our paper, as this corresponds to the point of view already given in Section 2 of our article.

l.86, The word 'tempted' is suggestive, consider: If you adopt the hand-to-hand mechanism such as Hansen [2019] and . . . . Be specific rather then suggestive.

We modified this portion to include an explicit citation of Hansen (2019), as this discussion paper supports this notion of the hand-to-hand mechanism **L292**:
*"If one adopts the hand-to-hand mechanism, such as Hansen (2019) for instance, the idea of water vapor shortcutting the ice may appear supported by the indistinguishability of water molecules."*

l.104. suggestion: 'and in particular if they are, on average, larger than . . . '

We modified the sentence to **L107**:

*"[...] and in particular if the macroscopic diffusion fluxes in snow are larger than the fluxes in free air."*

Consider including appendix A inside your main text. First it proves that D eff is maximal under infinitely fast kinetics, and is in fact equal to completely saturated condition (In the robin boundary condition such as used in Kaempfer and Plapp [2009] this limit is well defined). Then you proof that it is smaller than equation 8.

We fear that including Appendix A in the main text will break the flow of the article, and would prefer keeping the current structure.

The parameter $\phi$ is usually assigned to the ice volume fraction, it might therefore be helpful if you state that when you define it, for example: 'not to be confused with ', and restate that in your appendix when you re-derive the equations of Hansen and Foslien [2015].

We modified the article to clearly states that $\phi$ should not be confused with the ice volume fraction at the beginning of Section 4.1 (**L214**) and at the beginning of Appendix C (**L588**).

Appendix C: I did not have time to check all the volume fraction terms in comparison with Hansen and Foslien [2015], sorry for that. Maybe double check that there are no mistakes there. One question here: why is the tabular flux weighted by the ice phase and the laminar one by the vapor? Also it might be interesting to check what the origin is of this $\varphi$ difference term, is it only the hand-to-hand mechanism? or does it come from different definitions of averaging?

We decided to give more details on our derivation of the Hansen and Folsien (2015) model. The end result is the same, but our reasoning is now more clearly stated. Moreover, we now also discuss more clearly where the original error of the Hansen and Folsien (2015) model lies.

The modifications to the manuscript occur from **L596** to **L612**, with two added intermediate equations. As this section is rather long and contains mathematical equations, we did not copy it here. This modified section can be read page 24 of the manuscript attached at the end of the document, with all changes compared to the previous manuscript highlighted.

From our understanding the error of the original Hansen and Folsien (2015) article stems from how the laminar heat flux is decomposed into a purely conductive component and a latent heat flux component. Hansen and Folsien (2015) first rewrite the total heat flux under the form "A + L $dc_{sat}/dT$ B", and identify the term B with the diffusion coefficient. There are however multiple ways of decomposing the total heat flux under the aforementioned form, and the identification of $D_{eff}$ with B is therefore illegitimate.

The weighting of the tubular and laminar fluxes by the ice and vapor fractions is originally proposed in Hansen and Folsien (2015). It is not further discussed in our paper as our only goal is to demonstrate that their model does not produce vapor flux larger than in free air, and not to assess the validity of this model for snow modeling.

The weighting of the model is justified in Section 5.3 of the Hansen and Folsien (2015): *"heat transfer through the ice phase is dominated by the pore [tubular] microstructure where the thermal conductivity of ice is nearly 100 times that of air. In contrast, anytime the test line passes through the humid air constituent, heat transfer would be dominated by the lamellae microstructure."* Even though we are still not fully convinced by the proposed argument, we acknowledge that the Hansen and Folsien model predicts reasonable thermal conductivity values for a large range of density.

The brackets <, > look funny, rather use hci. (\left > and \left >).

We modified the brackets accordingly in Appendix A, from **L537** to **L566**.

word usage: preponderant? Use simpler words whenever possible. Consider deleting this sentence, since this is discussed in your paper, but is not a result of your paper.

We swapped the work preponderant with important **L11**:
*"Our results imply that processes other than diffusion play a predominant role in water vapor transport in dry snowpacks"*

The general word 'indeed' sounds very odd in a few places and is sometimes confusing, and rarely used in the beginning of a sentence. Please check the meaning of these sentences. For example l.494, you can avoid using Indeed, by: "Both Trabant and Benson (1972) and Sturm and Johnson (1991) already pointed out the importance of..."

We feel that the usage of "Indeed" at the start of sentences is appropriate as a conjunction, but we are aware that some may disagree on this point. We have thus reworded the sentences where indeed could be removed without altering the meaning or clarity. We have however kept the conjunction indeed **L362**, as we think it is the best way to introduce the justification for the preceding sentence.

l.7 – l.508 condensation is not replaced by deposition as you suggested you would do.

We are sorry to have left so many "condensation" words in our revised article, and should have double checked that point. We have now replaced the word "condensation" by "deposition". All modifications are highlighted in the track-change version attached at the end of this document.

last sentence of the conclusion: This is way to strong... you can't know that this is the only way, there might be other approaches that you haven't explored, please don't overstate.

We think it is important to stress out that in the case of non linear kinetics, it is no longer possible to compute the vapor flux by multiplying the concentration gradient by a constant diffusion coefficient (independent of the magnitude of the concentration gradient). It of course does not mean that it is no longer possible to derive a law relating the vapor flux to the concentration gradient, but that such law does not assume the classic proportionality form.

We have modified the last sentence of the conclusion to clarify this point **L526**:

[revised manuscript text omitted]
_{sat}\,\mathbf{n}\,dS = \left\langle \nabla c_{sat}\right\rangle - \nabla\left\langle c_{sat}\right\rangle \tag{A5}$$

Injecting Equation A5 in Equation A4 thus yields

$$\mathbf{F} = -D_0\nabla\left\langle c\right\rangle - D_0\left\langle \nabla c_{sat}\right\rangle + D_0\nabla\left\langle c_{sat}\right\rangle + \frac{D_0^2}{V\alpha v_{kin}}\int_{\Gamma}(\nabla c\cdot\mathbf{n})\,\mathbf{n}\,dS \tag{A6}$$

550 As we assume that the macroscopic vapor concentration equals the macroscopic saturation concentration gradient (as in Yosida et al. (1955); Colbeck (1993); Sokratov and Maeno (2000); Pinzer et al. (2012)), we have that $\nabla\langle c\rangle = \nabla\langle c_{sat}\rangle$. Thus

$$\mathbf{F} = -D_0\left\langle \nabla c_{sat}\right\rangle + \frac{D_0^2}{V\alpha v_{
[revised manuscript text omitted]